# 🐱 CHEMAGENT: SELF-UPDATING LIBRARY IN LARGE LANGUAGE MODELS IMPROVES CHEMICAL REASONING

**Xiangru Tang**[1,*]**Tianyu Hu**[1,*]**, Muyang Ye**[1,*]**, Yanjun Shao**[1,*]**, Xunjian Yin**[1],
**Siru Ouyang**[2]**, Wangchunshu Zhou, Pan Lu**[3]**, Zhuosheng Zhang**[4]**, Yilun Zhao**[1],
**Arman Cohan**[1]**, Mark Gerstein**[1]

[1]Yale University  [2]UIUC  [3]Stanford University  [4]Shanghai Jiao Tong University

xiangru.tang@yale.edu

## ABSTRACT

Chemical reasoning usually involves complex, multi-step processes that demand precise calculations, where even minor errors can lead to cascading failures. Furthermore, large language models (LLMs) encounter difficulties handling domain-specific formulas, executing reasoning steps accurately, and integrating code effectively when tackling chemical reasoning tasks. To address these challenges, we present ChemAgent, a novel framework designed to improve the performance of LLMs through a dynamic, self-updating library. This library is developed by decomposing chemical tasks into sub-tasks and compiling these sub-tasks into a structured collection that can be referenced for future queries. Then, when presented with a new problem, ChemAgent retrieves and refines pertinent information from the library, which we call memory, facilitating effective task decomposition and the generation of solutions. Our method designs three types of memory and a library-enhanced reasoning component, enabling LLMs to improve over time through experience. Experimental results on four chemical reasoning datasets from SciBench demonstrate that ChemAgent achieves performance gains of up to **46%** (GPT-4), significantly outperforming existing methods. Our findings suggest substantial potential for future applications, including tasks such as drug discovery and materials science. Our code can be found at https://github.com/gersteinlab/chemagent.

## 1 INTRODUCTION

Chemical reasoning presents unique challenges in the realm of artificial intelligence, demanding sophisticated reasoning and precise calculations beyond typical reasoning tasks (McQuarrie, 2008; Atkins et al., 2014; Talanquer, 2022; Guo et al., 2023b; Cao et al., 2024). For example, the GPT solution with CoT prompting in Figure 1 contains numerous errors in both the calculation process and the chemical constants used. Even in short reasoning chains, a single error can cascade, reducing answer quality and escalating the probability of additional errors (Liao et al., 2024; Sun et al., 2024b).

Recent advancements in large language models (LLMs) have demonstrated capabilities in simpler scientific tasks or chemical scenarios that do not require complex reasoning (Boiko et al., 2023; Atkins et al., 2023; Wang et al., 2024a; Hu & Shu, 2023; Xiao et al., 2024; Darvish et al., 2024; Skreta et al., 2024). Other approaches have investigated the impact of tool augmentation on chemistry problem-solving, with Yu et al. introducing ChemToolAgent. However, their application to complex chemical reasoning reveals significant limitations Pei et al. (2024); Li et al. (2024a). LLMs often (1) struggle to effectively utilize domain-specific formulas, (2) exhibit incorrect reasoning steps, and (3) produce errors when combining textual reasoning with Python code for calculations (Zhong et al., 2024b); here, syntax errors may arise, causing the code to fail to compile. As shown

---

*Equal contribution. The work was done when Mr. Hu and Mrs. Ye were interns at Yale University.

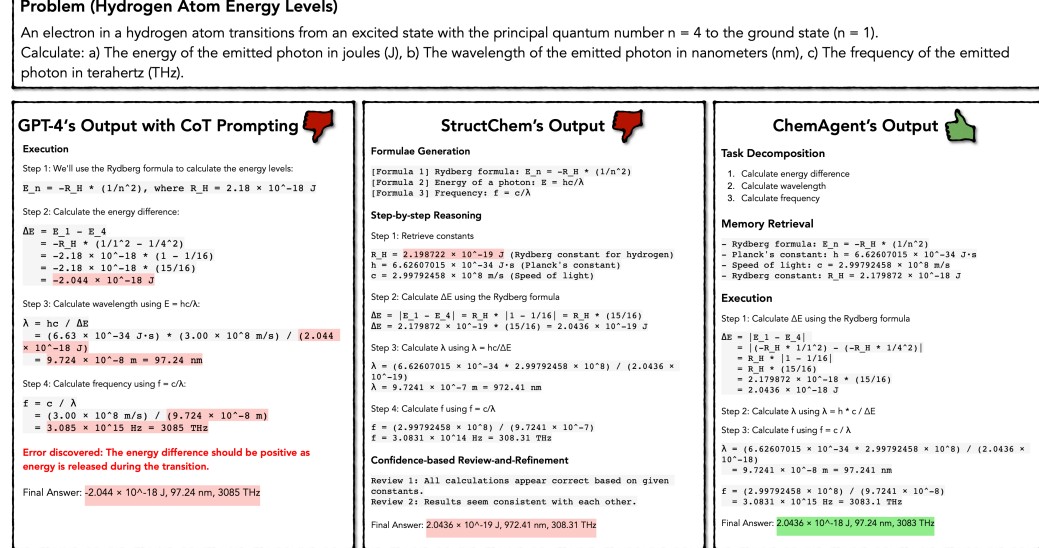

Figure 1: **Comparison of problem-solving approaches for a hydrogen atom energy transition problem.** The figure illustrates three different methods: **(a)** shows a standard Chain-of-Thought approach with calculation errors (in steps 3 and 4) in Wang et al. (2024a). **(b)** demonstrates the StructChem (Ouyang et al., 2024) method with formula generation and step-by-step reasoning but fails due to an incorrect constant and incorrect unit conversion (in steps 1 and 4). **(c)** presents the ChemAgent solution, featuring task decomposition, memory retrieval from the library, and reasoning, leading to the accurate final answer.

in Figure 1 (StructChem output), errors often arise from determining a constant's incorrect form or unit.

Some Previous approaches to address these challenges in chemical reasoning tasks have focused on decomposing the reasoning process (Zhong et al., 2023), e.g., formatting the reasoning steps through techniques such as self-reflection (Shinn et al., 2023) and StructChem prompting (Ouyang et al., 2024). However, these methods often rely heavily on human-curated knowledge (Chan et al., 2023), or fixed workflows (Fu et al., 2022; Wang et al., 2023; Zhou et al., 2023). Unlike human learners who utilize a library system to retain and apply previous experiences, these methods lack the ability to remember and learn from past analogous problems. For instance, humans can abstract and store theorems or solution strategies from previous tasks and utilize this memory for future problem-solving. [1]

Building on these insights, ChemAgent introduces a dynamic "LIBRARY" system that facilitates iterative problem-solving by continuously updating and refining its content based on task decomposition. The LIBRARY in ChemAgent serves as a comprehensive repository where decomposed chemical tasks are stored. These tasks are broken down into sub-tasks, and their solutions are compiled in the library for future reference. As new tasks are encountered, the library is updated with new sub-tasks and corresponding solutions, ensuring its content remains relevant and progressively improves over time. In parallel, inspired by human cognitive mechanisms (Osman, 2004; Kaufman, 2011), our system incorporates three main memory components: *Planning Memory* for high-level strategies, *Execution Memory* for specific task solutions, and *Knowledge Memory* for fundamental chemistry principles. These memories are stored externally, allowing the model to retrieve them efficiently during the problem-solving process. Unlike previous works relevant to LLM's external memory system (Huang et al., 2024a; Zhong et al., 2024a; Zhang et al., 2023; Li et al., 2024b; Guo et al., 2023a; Zhang et al., 2024b), we carefully integrate all these components in a complete agentic framework and allow them to update dynamically.

These memories are stored in a structured tree-based format that allows for efficient retrieval during the problem-solving process. When ChemAgent encounters a new problem, it first decomposes the task into manageable sub-tasks, then leverages the library to retrieve relevant sub-tasks and solutions stored in its memory components. The retrieved information is validated and refined using analogous sub-tasks previously encountered, optimizing both task decomposition and solution generation.

---

[1] For a more detailed discussion of the related work, please refer to Appendix A.

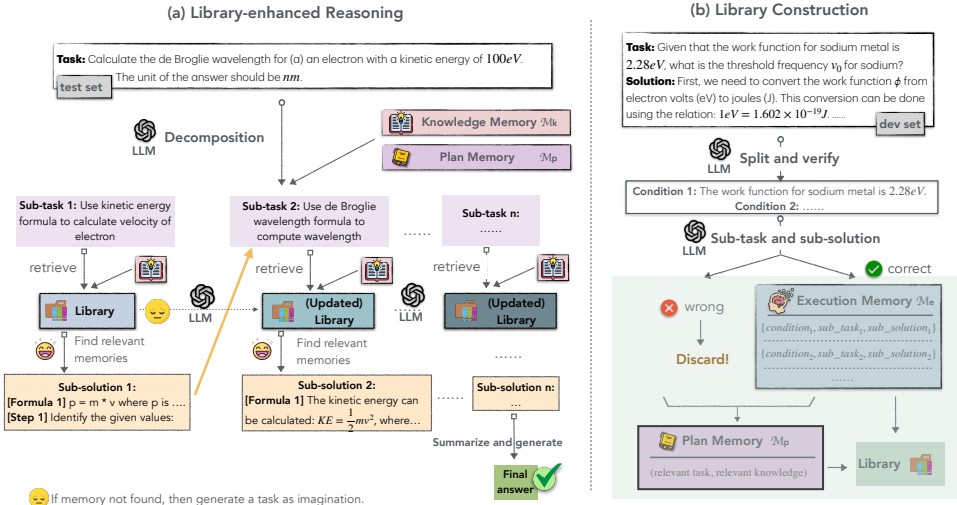

Figure 2: **The diagram of our overall framework.** It contains (a) library-enhanced reasoning and (b) library construction. **(a)** illustrates how ChemAgent utilizes the library to address a new task for the test set. And **(b)** demonstrates the construction of the library over the dev set, including Plan Memory $\mathcal{M}_p$ and Execution Memory $\mathcal{M}_e$).

The library is dynamically updated by adding new sub-tasks and solutions as they are encountered and validated. This iterative process ensures that the memory is continuously enriched with new strategies and solutions, mimicking human problem-solving improvements through practice. By maintaining an evolving domain knowledge base, ChemAgent enables LLMs to autonomously navigate the problem-solving process, thereby enhancing their performance on similar tasks over time.

Our experiments are conducted on four chemical reasoning datasets from SciBench (Wang et al., 2024a) with GPT-3.5, GPT-4 (OpenAI et al., 2024), and open-source models like Llama3 (Llama Team, 2024). Experimental results indicate that ChemAgent significantly enhances the accuracy, though the degree of improvement achieved by our model varies accordingly due to the varying sizes of these datasets. Compared to the base model (take GPT-4, for example), the application of memory self-improving leads to an average increase in accuracy by 37%, with a maximum improvement of 46%. In comparison with the current state-of-the-art model, StructChem (Ouyang et al., 2024), our approach achieves an average improvement of 10% and a maximum improvement of 15% across different datasets. Furthermore, the improvement is more pronounced for stronger base models: GPT-4 exhibits greater enhancements compared to GPT-3.5 when augmented with our framework. Additionally, we analyze the role of our library system and examine the benefits of different memory updates across various stages. This analysis provides insights into how our iterative library updates contributed to performance improvements in sub-task resolution and optimal task decomposition. By continuously enriching the library with structured memory components, ChemAgent ensures that the problem-solving process is progressively refined and optimized, leading to substantial improvements in accuracy.

## 2 METHOD

### 2.1 PRELIMINARIES

Analogous to how students organize and reference their problem-solving approaches for exams, our motivation for developing a library system in ChemAgent is to enhance LLMs' ability to tackle complex problems by providing structured access to a repository of previous sub-tasks and solutions.

The overall reasoning framework is shown in Figure 2 **(a)**. Formally, in a simplified form, given a complex and open-ended chemical problem $\mathcal{P}_\mathcal{S}$ as input, our method aims to generate a solution $\mathcal{O}_\mathcal{S}$. The problem $\mathcal{P}_\mathcal{S}$, which comprises problem descriptions $\mathcal{T}_\mathcal{S}$ and initial conditions $\mathcal{C}_\mathcal{S}$, is firstly decomposed into a series of sub-tasks $\mathcal{P}_{\mathcal{S}_i}$. The solution $\mathcal{O}_\mathcal{S}$ is then synthesized from the sub-solutions $\mathcal{O}_{\mathcal{S}_i}$ of the sub-tasks. Each $\mathcal{O}_{\mathcal{S}_i}$ includes intermediate steps, such as formulae, reasoning steps, code, and calculations. Then, a final answer $\mathcal{A}_\mathcal{S}$ is derived from the overall solution $\mathcal{O}_\mathcal{S}$. We evaluate the performance via the accuracy of $\mathcal{A}_\mathcal{S}$ against a ground truth $\mathcal{A}_{\mathcal{S}_g}$.

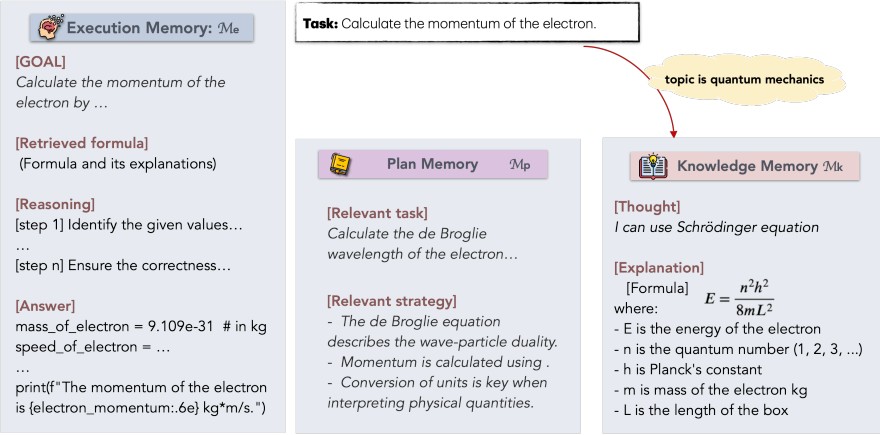

Figure 3: Given a task $\mathcal{P}$, the relevant memory examples are provided in the library. Specifically, while Execution Memory ($\mathcal{M}_e$) and Plan Memory ($\mathcal{M}_p$) are derived from prior experiences, Knowledge memory ($\mathcal{M}_k$) is generated by LLM based on the problem prompt. The conditions $\mathcal{C}$ are not explicitly presented here but are embedded within $\mathcal{P}$ and the [GOAL] of $\mathcal{M}_e$.

## 2.2 COMPOSITION OF THE LIBRARY

We divide *library* into three memory components: planning memory, execution memory, and knowledge memory. Figure 3 provides detailed examples, corresponding prompts are in Appendix G.

- **Planning Memory** ($\mathcal{M}_p$): This component stores high-level strategies and methodologies for approaching complex problems.
- **Execution Memory** ($\mathcal{M}_e$): This contains highly structured descriptions of specific problem contexts and their corresponding solutions, serving as detailed execution plans.
- **Knowledge Memory** ($\mathcal{M}_k$): This houses fundamental chemistry principles and formulas, acting as a ready reference. This component is generated temporarily during the solution of a specific task and is not intended for permanent retention.

Thus, ChemAgent enables LLMs to tackle problems proactively, form new memories from these attempts, utilize existing memories to solve complex problems, and continuously refine their solutions. Built upon the *library* of three types of memories, ChemAgent operates in two main stages:

1. **Initial Library Construction:** Complex problems in the development set are decomposed into structured atomic sub-tasks, which serve as building blocks of the library, consisting of static long-term memory ($\mathcal{M}_p$ and $\mathcal{M}_e$).
2. **Inference Phase and Dynamic Memory Updating:**
   (a) `Memory-Enhanced Reasoning:` The memory is dynamically enriched and improved during runtime.
   (b) `Evaluation & refinement:` Each sub-task's solution is evaluated on their veracity and relevance to the main task, which leads to a correction of the solution or a refinement of the overall plan.

Figure 2 and Figure 4 illustrates this workflow. In the following sections, we explain how each part is constructed and utilized. All prompts used in our framework can be found in Appendix G.

## 2.3 DECOMPOSITION AS ATOMIC BLOCKS

Human problem-solvers naturally break down complex chemistry problems into smaller, manageable sub-tasks (Zhou et al., 2023). This decomposition enhances the understanding of individual components and their interactions, facilitating the resolution of the original problem in a structured manner (Johnson et al., 2017). These sub-tasks not only improve the reasoning process but also function as atomic building blocks within the execution memory, each paired with a corresponding sub-solution. Key characteristics of this decomposition include:

**Hierarchical Decomposition**: Using the strategy stored in Planning Memory, we break down complex chemistry problems into hierarchical sub-tasks. If a sub-task is too difficult to complete, it is further decomposed until it can be executed in a single step by the LLM. We store each decomposed

sub-task and its intermediate sub-solutions, enabling direct retrieval of solutions for similar future problems. This hierarchical approach ensures each sub-task acts as an independent atomic block, aiding future problem-solving.

**Structured Sub-tasks**: To maximize the efficiency of decomposition, sub-tasks are structured into four distinct parts: (i) *Task Query* defining the specific question, (ii) *Sub-task Objectives* outlining clear goals, (iii) *Step-by-step Solution* detailing the method to address the sub-task, and (iv) *Guidance for Solutions* suggesting potential approaches.

**Atomic Blocks as Memory**: Decomposed sub-tasks serve as building blocks for memory. Identifying and recalling similar problems can be challenging, but sub-tasks often share commonalities that make it easier to leverage past experiences. These atomic blocks also function as examples in few-shot prompting, standardizing response formats and bolstering the overall problem-solving process.

Building on the concept of sub-tasks as fundamental units, we outline the processes involved in our framework, which includes both a dynamic library system and the structured memory inside our library. This section details how we initially construct the library and memory (§2.4), utilize and update them for new tasks (§2.5), and evaluate and refine the generated solutions (§2.6). Specifically, our approach leverages the development set to build a library, which is designed to dynamically self-improve during runtime.

## 2.4 LIBRARY CONSTRUCTION

The Library is constructed using the development set. As outlined in §2.3, we leverage sub-tasks decomposed from each problem as the execution memory units. Each execution memory unit $\mathcal{U}_i$ is defined as follows, where $\mathcal{C}$ represents the conditions of a given problem $\mathcal{P}$, and $\mathcal{T}_i$ and $\mathcal{O}_i$ denote the sub-task and its corresponding sub-solution, respectively:

$$\mathcal{U}_i = (\mathcal{C}, \mathcal{T}_i, \mathcal{O}_i) \quad \text{for} \quad i = 1, 2, \ldots, k$$

Given a problem $\mathcal{P}$ and its corresponding solution $\mathcal{S}$ in the development set, our method begins by identifying and extracting the conditions from $\mathcal{P}$. We then verify these conditions for accuracy to ensure that the subsequent steps operate with precise and correctly parsed data. Based on the identified conditions, we instruct the LLMs to generate detailed

---

**Algorithm 1:** Library Construction

**Input:** Development set $\mathcal{D}$, LLM $\mathcal{F}$,
      prompts $\{p_{\text{split}}, p_{\text{ref}}, p_{\text{rank}}\}$
**Output:** Static memory $\mathcal{M}$ consisting of
      units
      $\mathcal{U} = \{\text{condition}, \text{question}, \text{solution}\}$
**for** $(\mathcal{P}, \mathcal{S})$ *in* $\mathcal{D}$ **do**
    Conditions $\mathcal{C} \leftarrow \mathcal{F}(p_{\text{split}} \| \mathcal{P})$
    // Verify conditions and
      refine if necessary
    $\mathcal{C} \leftarrow \mathcal{F}(p_{\text{ref}} \| \mathcal{P} \| \mathcal{C})$
    Sub-tasks $\mathcal{T} \leftarrow \mathcal{F}(p_{\text{task}} \| \mathcal{P} \| \mathcal{C})$
    Sub-solutions $\mathcal{O} \leftarrow \mathcal{F}(p_{\text{sol}} \| \mathcal{C} \| \mathcal{P} \| \mathcal{S})$
    **Assert** $len(\mathcal{C}) = len(\mathcal{T}) = len(\mathcal{O})$
    **for** $i$ *in* $len(\mathcal{C})$ **do**
        Add $(\mathcal{C}_i, \mathcal{T}_i, \mathcal{O}_i)$ into $\mathcal{U}$
    $\mathcal{U} \leftarrow \mathcal{F}(p_{\text{rank}} \| \mathcal{U})$
**return** $\mathcal{U}$

---

sub-tasks. For each identified sub-task $\mathcal{T}_i$, the corresponding sub-solutions $\mathcal{O}_i$ are parsed from $\mathcal{S}$ and assigned accordingly. Inspired by curriculum learning (Bengio et al., 2009), we then rank the memory units $\mathcal{U}$ based on their difficulty. In addition to ranking, we discard any memory units that do not meet a predefined confidence threshold, as evaluated by the LLMs. This ensures that the memory utilized for future problem-solving is both relevant and reliable, enhancing the LLM's ability to tackle increasingly complex chemistry problems. The detailed memory construction process is further described in Algorithm 1.

## 2.5 LIBRARY-ENHANCED REASONING

During testing, we first decompose a given problem into several sub-tasks. For each sub-task, we retrieve related memory units $\mathcal{U}_r$ to aid in solving it. Specifically, we compute the similarity between the given sub-task and units stored in the memory. Memory units with similarity above a predefined threshold $\theta$ are used to assist the model in determining the answer for the sub-task.

Formally, let $(\mathcal{C}_j, \mathcal{T}_j)$ represent a sub-task decomposed from a new problem. We retrieve memory units $\mathcal{U}_r$ that satisfy

$$\text{Similarity}(\mathcal{T}_j, \mathcal{T}_{\mathcal{U}_i}) \geq \theta \quad \text{for} \quad \mathcal{U}_i \in \mathcal{M}_e.$$

Specifically, the similarity between tasks is calculated using Llama3's embeddings:

$$\text{Similarity}(T_a, T_b) = \frac{\text{Embed}(T_a) \cdot \text{Embed}(T_b)}{||\text{Embed}(T_a)|| \times ||\text{Embed}(T_b)||}.$$

In scenarios where the *execution memory* does not contain similar sub-tasks, we incorporate a self-improving mechanism. Initially, we enrich the memory with the required information by leveraging the internal parametric knowledge of LLMs. The LLM is instructed to identify the *topic* of the given sub-task (e.g., quantum chemistry or thermodynamics) and generate self-created chemistry problems related to that topic, adhering to specific guidelines. This forms a kind of "synthetic" execution memory. We encourage the LLMs to generate diverse problems and solutions beyond the provided examples, thereby enhancing the robustness and adaptability of the memory system.

Moreover, the memory is continuously updated with newly solved sub-tasks and their solutions during runtime. For each newly solved sub-task $\mathcal{T}_j$, sub-answers from previous sub-problems are incorporated into the conditions as $\mathcal{C}_j$, and the execution memory is updated based on solution $\mathcal{O}_j$:

$$\mathcal{M}_e = \mathcal{M}_e \cup \{(\mathcal{C}_j, \mathcal{T}_j, \mathcal{O}_j)\}.$$

The plan memory will be updated by a summary of the overall strategy knowledge used, $\mathcal{K}_j$ (e.g. formulas, conception, order of resolution):

$$\mathcal{M}_p = \mathcal{M}_p \cup \{(\mathcal{T}_j, \mathcal{K}_j)\}.$$

This dynamic updating ensures that the memory evolves and improves over time, progressively enhancing the problem-solving capabilities. More details on self-evolution are in §2.9.

## 2.6 EVALUATE & REFINE MODULE

To enhance the flexibility and reliability of ChemAgent, we propose an `evaluation & refinement` module, to correct the planning trajectory and verify the response to each sub-task.

As illustrated in Figure 4, after addressing a specific sub-task $\mathcal{P}_i$, ChemAgent examines the sub-solution $\mathcal{O}_i$. The system evaluates whether the sub-solution conflicts with fundamental knowledge in $\mathcal{M}_k$ or contains common errors, such as incorrect units. If discrepancies are identified, a new sub-solution $\mathcal{O}'_i$ is generated by refining the original $\mathcal{O}_i$ based on the relevant knowledge in $\mathcal{M}_k$ and the identified mistakes.

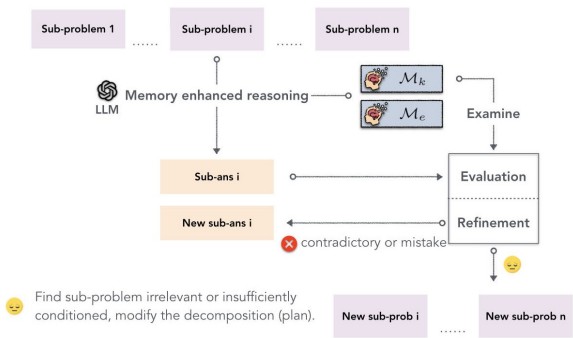

Figure 4: **Overall framework of the `evaluation & refinement` module.** ChemAgent continuously modifies the solution or the comprehensive strategy until it either reaches the maximum number of trials or meets the evaluator's criteria.

Furthermore, if a sub-task fails due to insufficient conditions or if the evaluation determines that the sub-task's question does not align with the main task (e.g., calculating energy using wavelength instead of calculating wavelength using given energy), the sub-tasks from $\mathcal{P}_i$ to $\mathcal{P}_n$ will be restructured as $\mathcal{P}'_i$ to $\mathcal{P}'_m$, taking into account the main task and all preceding sub-tasks.

The `evaluation & refinement` module can use a different LLM than the one used for the base reasoning process. For instance, if GPT-3.5 is the base model, GPT-4 can handle `evaluation & refinement`. The evaluation component judges solutions without modifying them, while the refinement component adjusts solutions based on these evaluations. This separation clarifies error identification, helping humans understand where and why mistakes occur.

## 2.7 SETUP

We use four chemistry datasets from SciBench (Wang et al., 2024a), and the detailed distribution of the specific fields covered by each problem in the four datasets is shown in Figure 10. Each dataset is divided into a development set ($\mathcal{D}_d$) and a test set ($\mathcal{D}_t$), with exact sizes provided in Table 6. According to previous research (Ouyang et al., 2024), SciBench chemical sets are representative of chemical reasoning tasks, addressing a broad range of queries typically encountered in this field.

During the reasoning stage, we configure the planning memory ($\mathcal{M}_p$) to provide a maximum of two related memory instances (2-shot) for each query, and the execution memory ($\mathcal{M}_e$) to provide up to four related instances (4-shot). However, during the construction of the library, only the knowledge

Table 1: Performance on the test sets of four datasets: QUAN, CHEMMC, ATKINS, and MATTER. Results are compared with baselines under two different setups: a zero-shot setting with no demonstrations and a few-shot setting with three demonstrations. Accuracy scores are computed using the approximation detailed in Section 4.3. The best results for each setup are highlighted in **bold**, and the second-best results are underlined. In some settings, the two modules described in Sections 2.4 (Memory) and 2.6 (Evaluation & refinement) are disabled, helping illustrate each module's impact on the overall performance.

| Models | CHEMMC | MATTER | ATKINS | QUAN | Avg. |
|---|---|---|---|---|---|
| *Baselines, all based on GPT-4 (gpt-4-1106-preview)* | | | | | |
| Few-shot + Direct reasoning | 28.21 | 14.29 | 20.69 | 14.71 | 19.48 |
| Few-shot + Python | 38.46 | 34.69 | 57.01 | **44.12** | 43.57 |
| StructChem | 58.97 | 30.67 | 59.81 | 41.18 | 47.66 |
| *ChemAgent, all based on GPT-4 (gpt-4-1106-preview)* | | | | | |
| ChemAgent | **74.36** | **48.98** | **61.18** | **44.12** | **57.16** |
| w/o Evaluation & Refinement | 61.54 | 44.89 | 57.94 | **44.12** | 52.12 |
| w/o Memory, Evaluation & Refinement | 58.97 | 38.78 | 57.94 | 41.18 | 49.22 |
| *all based on Llama 3.1-7b* | | | | | |
| Direct reasoning | 28.20 | 10.20 | **22.43** | 8.82 | 17.41 |
| ChemAgent | **56.41** | **12.24** | 19.63 | **14.71** | 25.75 |
| *all based on Llama 3.1-70b* | | | | | |
| Direct reasoning | 33.33 | 30.61 | 30.84 | 23.53 | 29.48 |
| ChemAgent | **64.10** | **32.65** | **43.93** | **29.41** | **42.52** |
| *all based on Qwen 2.5-72b* | | | | | |
| Direct reasoning | 48.72 | 32.65 | **57.01** | 35.50 | 43.47 |
| ChemAgent | **69.23** | **44.90** | 56.07 | **44.12** | **53.58** |

memory ($\mathcal{M}_k$) is used, as the standard solutions are already available in the development set ($\mathcal{D}_d$). We evaluate the accuracy by comparing their outputs with the correct answers, using a relative tolerance of 0.01.

We consider three baselines in alignment with the evaluation paradigm in SciBench (detailed instructions are provided in Appendix G): **(1) Few-shot + Direct reasoning** involves directly feeding the problem into the LLM without any additional instructions, the source of the data is StructChem (Ouyang et al., 2024). **(2) Few-shot + Python** combines few-shot examples with a tool-augmented strategy. Here, we provide six examples for every query, each consisting of a problem, its corresponding solution, and Python code. These examples are taken from $\mathcal{D}_d$. The results are from the original benchmark, SciBench (Wang et al., 2024a). **(3) StructChem** (Ouyang et al., 2024) employs structured instruction, confidence-based review, and refinement to guide LLMs through precise step-by-step reasoning.

## 2.8 RESULTS

We report the performance of all methods regarding accuracy score for each sub-dataset and an average score across the entire dataset. The results are summarized in Tables 1, 4 and 5. Additional results and analysis of experiments conducted on other LLMs can be found in Appendix B.

Firstly, ChemAgent consistently outperforms the other baselines across various datasets and settings. Specifically, in terms of the average score, ChemAgent improves by **9.50%** (47.66 vs. 57.16) over StructChem, which is a 2.93 times increase and by **37%** (19.48 vs. 57.16) over direct reasoning, which is a 2.93 times increase. Notably, the performance gain varies across different datasets. In the CHEMMC dataset, our method exhibits the largest improvement, with a **46%** increase (28.21 vs. 74.36) compared to the direct reasoning setup. Secondly, the results also highlight the crucial role of memory in our library. The version of our framework equipped with memory consistently outperforms the version without memory across all cases. Specifically, there is an absolute improvement of **2.90%** in terms of the average score. This underscores the importance of memory in retaining and leveraging past information to improve the accuracy and reliability of solving chemistry problems. Additionally, the Evaluation & Refinement module plays a significant role when using stronger LLMs. For instance, incorporating Evaluation & Refinement with GPT-4 increases ChemAgent's performance by **5.04%** compared to ChemAgent with memory alone. In summary, ChemAgent demonstrates superior performance across various LLM backbones, highlighting the critical importance of the library system, memory design, and evaluation modules in

enhancing problem-solving capabilities. Appendix C details the specific aspects and reasons for our method's improved performance.

## 2.9 SELF-EVOLUTION DURING RUNTIME

Moreover, we aim to show that library systems with evolving memory perform better when exposed to an increasing number of problems. Much like humans improve their skills through practice, these systems benefit from continuous exposure to new tasks.

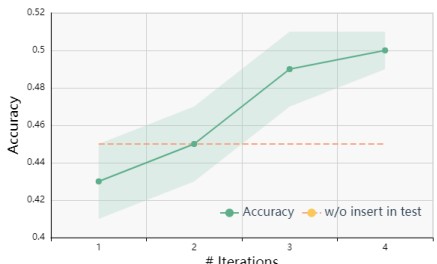

We allow ChemAgent to dynamically update and enrich its library during the test stage to analyze this self-evolution process. Specifically, in iteration $\mathcal{I}_i$, ChemAgent uses all accumulated long-term memory from iterations $\mathcal{I}_1$ to $\mathcal{I}_{i-1}$ as its library. When solving a new problem $\mathcal{P}$, all related responses and knowledge from that process are added to the library if the solution is correct. This means that in subsequent iterations, the system can leverage the newly acquired information (updated $\mathcal{M}_p$ and $\mathcal{M}_e$) to improve its performance. We employ 2-shot $\mathcal{M}_p$ and

Figure 5: **Self-evolving analysis.** We test ChemAgent twice for each iteration, and the difference between the two results serves as the error margin. All the experiments here are done on `MATTER` dataset.

4-shot $\mathcal{M}_e$ during reasoning but simplify the setup by removing the evaluation and refinement modules. To ensure accuracy and prevent target leakage, the memory derived from problem $\mathcal{P}_i$ in iteration $\mathcal{I}_j$ is excluded when solving $\mathcal{P}_i$ again in $\mathcal{I}_k$ for any $j < k$.

This iterative process demonstrates that as the memory pool grows with each new example, ChemAgent's problem-solving performance improves. Figure 5 shows that as the number of iterations increases, the agent's performance gradually improves and converges to a score higher than the baseline (44.89%). This improvement indicates that ChemAgent can enhance its performance through a simple correct-or-not evaluation of past solutions instead of human-written high-quality trajectories.

## 2.10 COST ANALYSIS

On average, each problem requires 0.012 million tokens without the `Evaluation & Refinement` module, equivalent to around $0.09 per example. When this module is applied, the average token consumption per problem increases to approximately 0.023 million, costing about $0.1725 per example. Note that the initial library construction is not included in these calculations as it is a one-time setup. Based on the results in Figure 6, the computational time required by our method is reasonable. While the cost and resource consumption are slightly higher than StructChem, the improvements justify the additional expense.

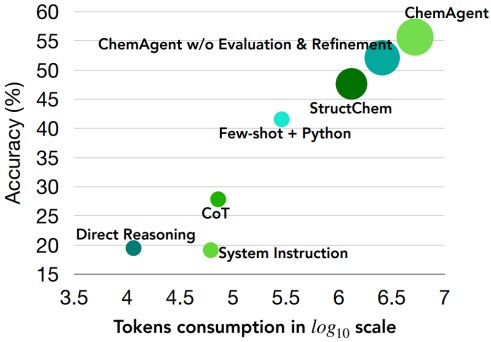

Figure 6: **Cost Analysis.** The size of each bubble corresponds to the average number of inferences for each method, while the *y-axis* indicates the average accuracy across the four datasets.

## 2.11 ERROR ANALYSIS

We conduct an analysis of the trajectories for the failed examples and find three types of errors, shown in Figure 7.

**(1) Lack of Understanding of the Question.** We observe that plans often fail when the problem text contains critical hidden information (e.g., in Figure 7, the terms 'reversibly' and 'adiabatically' are key) or include excessive, redundant details (e.g., the exact conditions such as "250K" are irrelevant to the solution). This challenge is understandable, as even human solvers can be misled by such information and err in their approach. This issue is prevalent and independent of whether the task is tackled by our model or directly queried to the LLM, suggesting that these errors may be inherent limitations of the LLM's capacity. Hence, improving the ability of the foundation model to handle such nuances could significantly enhance performance.

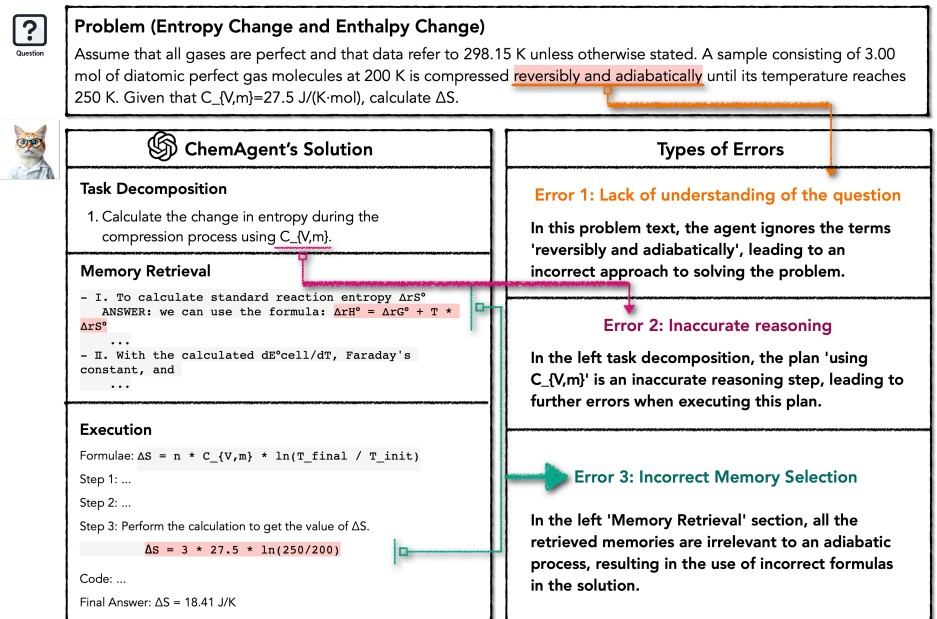

Figure 7: **Error analysis.** This example highlights a typical incorrect solution, which can be attributed to three main types of errors. Specifically, this problem pertains to an adiabatic process.

**(2) Inaccurate Reasoning.** As indicated before (Ouyang et al., 2024), the planning capabilities of LLMs remain insufficient for handling complex problems. Incorrect planning for the reasoning chains exacerbates the problem-solving process, as subsequent decisions and actions are based on initial problem decompositions. This issue persists until the `Evaluation & Refinement` module detects an error, which may not happen promptly enough to correct the trajectory.

**(3) Incorrect Memory Selection.** While ChemAgent with memory demonstrates superior performance compared to setups without memory, it sometimes invokes misleading information, even when the similarity between the invoked memory and the problem is high. This indicates a need for more sophisticated memory retrieval and utilization strategies.

In Figure 7, the invoked memory and sub-task 1.1 share considerable similarities—both concern entropy change during a compression process. However, the subtle difference is that the current problem involves an adiabatic process, whereas the examples in memory do not. This seemingly minor distinction can lead to significant changes in the problem-solving strategy. Distinguishing between misleading and beneficial memories remains a challenging issue, as invoked memories and problem texts may be semantically similar yet differ in critical aspects.

## 3 ABLATION STUDY

### 3.1 MEMORY COMPONENT ANALYSIS

To understand why our method performs particularly well and which memory component contributes the most, we conduct an ablation study by independently removing each memory component. We test these different settings on all four sub-datasets using GPT-4 as the base LLM. The results are shown in Table 2. As mentioned in §2.3, the execution memory also serves as a few-shot prompting strategy. Therefore, when we remove $\mathcal{M}_e$, we add two fixed human-written few-shot examples (provided with our code) into each query of each sub-task. These examples are selected from the development set ($\mathcal{D}_d$) corresponding to each dataset to ensure they do not provide misleading information.

Firstly, removing any memory component results in an overall performance drop. And notably, the relevant knowledge in LLM itself ($\mathcal{M}_k$) significantly impacts the overall performance of ChemAgent. Delving into why $\mathcal{M}_k$ contributes the most, we find that this might be due to the insufficient $\mathcal{M}_p$ and $\mathcal{M}_e$ in the Atkins dataset. As shown in Table 6, ATKINS has the lowest ratio of the development set to the test set, while it also has the largest test set. This imbalance may result in a small and sometimes irrelevant memory pool for $\mathcal{M}_p$ and $\mathcal{M}_e$ to search from. We hypothesize that this issue can be mitigated when the model is exposed to more chemical questions over time. However, due to limited computational resources, we cannot scale up to verify this hypothesis in this research.

Table 2: The best performing score is highlighted in **bold** and second-best is underlined. The number of problems in each textbook weighs the average score. We also provide the results of ChemAgent without any of these memories as a baseline. `Evaluation & refinement` module is removed.

| Memory component | CHEMMC | MATTER | ATKINS | QUAN | Avg. |
|---|---|---|---|---|---|
| None | 58.97 | 38.78 | 57.94 | 41.18 | 51.53 |
| $+ \mathcal{M}_p\ \mathcal{M}_e\ \mathcal{M}_k$ | 61.54 | 44.89 | 57.94 | **44.12** | **53.71** |
| $+ \mathcal{M}_p\ \mathcal{M}_e$ | 66.66 | 44.89 | 52.34 | 41.18 | 51.53 |
| $+ \mathcal{M}_e\ \mathcal{M}_k$ | 58.97 | **46.94** | **61.68** | 29.41 | 53.27 |
| $+ \mathcal{M}_p\ \mathcal{M}_k$ few-shot | **74.36** | 42.86 | 57.94 | 32.35 | **53.71** |

It is also worth noting that although the $M_p, M_k, fewshot$ setting achieves the same accuracy as the complete memory setting, most of its improvement is evident in the CHEMMC dataset, showing surprisingly higher improvements than others. Since Table 6 and Figure 10 demonstrate that the CHEMMC dataset has a higher ratio of $\mathcal{D}_d$ to $\mathcal{D}_t$ and a relatively narrow distribution in subfields, and since human experts carefully wrote the few-shot examples, these high-quality, expert-written examples may serve as a superior memory pool in such circumstances. In the next section, we show that better memory leads to better results, which might explain this phenomenon.

## 3.2 MEMORY QUALITY INFLUENCE

We investigate the impact of memory quality through a series of experiments. Specifically, we compare the performance of memory generated by GPT-4 against that generated by GPT-3.5 on the MATTER dataset. Additionally, we create a "hybrid memory" by mixing memories generated by both GPT-3.5 and GPT-4 to observe its performance. These experiments were conducted without including the relevant knowledge within the LLM itself ($\mathcal{M}_k$). As mentioned in §2.1, unlike $\mathcal{M}_p$ and $\mathcal{M}_e$, $\mathcal{M}_k$

Table 3: Results of memory quality analysis, with the best scores highlighting in **bold**. Here, memory refers to $\mathcal{M}_p$ and $\mathcal{M}_e$. The relevant knowledge in LLM itself ($\mathcal{M}_k$) is removed from the framework. Test on MATTER dataset.

| | GPT-3.5 | GPT-4 |
|---|---|---|
| GPT-3.5 Memory | 11.22 | 36.73 |
| GPT-4 Memory | **13.26** | **44.89** |
| Hybrid Memory | 10.20 | 28.57 |

is not preserved in the memory pool shared with other LLMs. Including this type of memory in the ablation study would result in the pollution of provided memory by the $\mathcal{M}_k$ generated during use.

As shown in Table 3, our data analysis indicates that memory quality significantly impacts the accuracy of responses from ChemAgent. Whether GPT-3.5 or GPT-4 is used, the memory produced by GPT-4 consistently outperforms that generated by GPT-3.5, showing an 8% variation. This underscores the substantial impact of memory quality on the accuracy of generated responses. This observation also explains why the improvements seen with GPT-3.5 are less pronounced compared to those with GPT-4; the lower quality of memory produced by GPT-3.5 is the limiting factor.

Interestingly, hybrid memory performs the worst among the three settings. This can be attributed to the simultaneous invocation of different memories for the same question—one generated by GPT-3.5 and the other by GPT-4—which may confuse the LLM, increasing the likelihood of producing irrelevant or incorrect answers. Additionally, invoking an excessive number of memory tracks during resolution may also contribute to this issue. A more detailed analysis regarding the number of memory tracks invoked is provided in Appendix D.

## 4 CONCLUSION

Our research presents a novel approach to enhancing large language models for solving complex chemical problems through self-exploration and memory formation. This method enables models to construct and utilize a library, significantly improving response accuracy. Experiments using datasets and models like GPT-3.5, GPT-4, and Llama3 demonstrate substantial performance gains, with the ChemAgent architecture achieving up to a 36% improvement. The structured library, built through memory decomposition into planning, execution, and knowledge, enhances problem-solving capabilities, which holds promise for generalization to other domains.

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

# A   RELATED WORKS

## A.1   CHEMICAL REASONING

Recent advances in LLMs have shown promise in scientific reasoning, yet chemical reasoning remains particularly challenging. Benchmarks like SciBench (Wang et al., 2024a) have revealed that current LLMs struggle significantly with complex chemical calculations and multi-step reasoning tasks. SciBench includes 869 college-level problems across mathematics, chemistry, and physics, providing a rigorous evaluation of LLM capabilities in these domains. Other datasets (Wadden et al., 2024; Li et al., 2024c; Sun et al., 2024a; Feng et al., 2024; Huang et al., 2024b) have also contributed to advancing the evaluation of LLMs in scientific problem-solving.

In response to these challenges, several approaches have been proposed. StructChem (Ouyang et al., 2024) provides structured guidance by decomposing chemical reasoning into phases such as formula generation, detailed step-by-step reasoning, and confidence-based review. While showing improvements, it still faces limitations with highly complex problems. Other researchers have explored enhancing LLM performance through various prompting strategies (Yang et al., 2024; Yao et al., 2024; Besta et al., 2024).

Some studies have focused on using Python code for reasoning tasks. Jie et al. (2023) demonstrated that self-describing programs, which transform reasoning processes into executable code, can significantly outperform natural language Chain-of-Thought methods (Wei et al., 2022) in certain scenarios.

Specialized tools like ChemCrow (Bran et al., 2023) utilize function calling and precise code generation to tackle chemistry problems. Davis & Aaronson (2023) highlight that combining GPT-4 with external tools like Wolfram Alpha or Code Interpreter can significantly improve problem-solving in science. However, reliably integrating these tools for complex chemical calculations remains a challenge.

Yu et al. proposed ChemToolAgent, a comprehensive tool-augmented framework for chemistry, demonstrating that while specialized tools significantly boost performance on molecular manipulation tasks, they may introduce additional complexity that hinders LLMs' reasoning abilities on general chemistry questions that require fundamental knowledge.

Compared to those tool-augmented approaches, which excels in specialized molecular tasks but struggles with complex reasoning in general chemistry questions, our ChemAgent demonstrates consistent improvements across both specialized and general chemistry reasoning tasks through its self-updating library mechanism.

## A.2   PROBLEM DECOMPOSITION IN SCIENTIFIC REASONING

Decomposing complex problems into smaller sub-tasks has shown to enhance model understanding and accuracy across various domains. In the context of chemical reasoning, this approach is particularly relevant due to the multi-step nature of many chemical problems. Patel et al. (2022) highlights the efficacy of question decomposition by breaking down complex queries into manageable sub-tasks. Similarly, Khot et al. (2022) demonstrates the benefits of modular task decomposition. Other studies (Lu et al., 2022; Wei et al., 2023) further underscore the advantages of decomposition in complex question answering and reading comprehension. Our work builds on these insights, specifically focusing on how breaking down complex chemical problems can improve the performance of self-evolving agents in this domain.

## A.3   SELF-EVOLUTION AND SELF-CORRECTION IN AGENT REASONING

Recent research has explored the self-evolution and optimization of LLMs, which is particularly relevant for tackling the complexities of chemical reasoning. Yang et al. (2023) explore methods for enhancing LLM performance through self-improvement techniques, while Fernando et al. (2023) investigate self-referential self-improvement via prompt evolution. Additionally, Zhou et al. (2024), Jiang et al. (2023), Hu et al. (2024) and Qian et al. (2024) present frameworks for agent self-evolution, aligning with our approach of enabling self-exploration and continuous learning in complex chemical problem-solving. While some of these frameworks also incorporate a memory

system, they predominantly emphasize the reuse of past workflows in daily tasks, as demonstrated by Wang et al. (2024b) and Qian et al. (2024).

Furthermore, in chemical reasoning, where even minor errors can lead to significant discrepancies in results, the ability of LLMs to self-correct and refine their outputs is crucial. Huang et al. (2023) investigate the limitations of self-correcting mechanisms in LLMs, while Weng et al. (2022) explore the potential of LLMs as reasoners with self-verification capabilities. In terms of tool-assisted methods, Gou et al. (2023) propose a framework for LLMs to self-correct with tool-interactive critiquing, and Gao et al. (2023) offer insights into researching and revising LLM outputs using LLMs themselves. Other studies like Qu et al. (2024) also inform our approach to developing a self-updating memory system capable of refining its chemical reasoning processes.

## B    EXPERIMENTAL RESULTS OF MODELS OTHER THAN GPT-4

We also evaluated ChemAgent thoroughly on earlier and less powerful models, such as GPT-3.5 (`gpt-3.5-turbo-16k`). Specifically, ChemAgent achieves an absolute improvement of +0.17% in terms of the average score compared with the previous SOTA. Also, our framework significantly performs better when equipped with a memory system (+7.13% absolute improvement), which underscores the importance of memory.

However, when it comes to the evaluation and refinement module, an interesting phenomenon is observed: even when outputs generated by GPT-3.5 are evaluated by more capable models like GPT-4, the agent often fails to correct its mistakes. This indicates that GPT-3.5 has a relatively weaker self-correction ability than GPT-4, explaining why evaluation and refinement provide little benefit when GPT-3.5 is used. This finding aligns with previous research (Zhang et al., 2024a).

Table 4: Performance on the test sets of four datasets: `QUAN`, `CHEMMC`, `ATKINS`, and `MATTER`. We compare baselines under two different settings: a zero-shot setting with no demonstrations and a few-shot setting with three demonstrations. The accuracy scores are computed with the approximation detailed in Section 4.3. The best results under each setup are highlighted in **bold**, and the second-best results are underlined. This experiment is done using GPT-3.5 (`gpt-3.5-turbo-16k`)

| GPT-3.5 | CHEMMC | MATTER | ATKINS | QUAN | Avg. |
|---|---|---|---|---|---|
| *Baselines, all based on GPT-3.5* | | | | | |
| Few-shot+Direct reasoning | 23.08 | 8.16 | 9.35 | 5.88 | 11.61 |
| Fewshot+Python | 33.33 | 16.33 | 13.08 | 8.82 | 17.89 |
| StructChem | 43.59 | **24.49** | 26.17 | **32.35** | 31.65 |
| *ChemAgent, all based on GPT-3.5* | | | | | |
| ChemAgent w/o memory | 38.46 | 16.33 | 23.36 | 20.59 | 24.69 |
| ChemAgent w/ memory | **56.41** | 18.37 | **28.97** | 23.53 | **31.82** |
| + evaluate and refine | 41.03 | 16.33 | 28.04 | 20.59 | 26.50 |
| + evaluate by GPT-4 and refine | 46.15 | 22.45 | **28.97** | 26.47 | 31.01 |

For open-source models, we choose Llama3-7b-Instruct, Llama3-70b-Instruct, and Qwen2.5-72b-Instruct as the experimental model here. The baseline direct reasoning is to directly query the model without adding other instructions. The evaluation and refinement modules are removed from the ChemAgent configuration, and only the Execution Memory ($\mathcal{M}_e$) is added due to the model's relatively lower ability on instruction following. Each experiment is repeated at least 3 times, and the results are averaged. On llama3-7b, the average increase across four datasets is 8.34%. On llama3-70b, the average increase across four datasets is 13.04%. The experiment demonstrates that the stronger the self-capabilities of large models, the more pronounced the performance gains using our framework.

Reference to Appendix F, the *atkins* dataset, and the *matter* dataset involve a lot of specialized knowledge in chemistry. The questions in *atkins* involve the theoretical aspects of thermodynamics, kinetics, and other chemical processes, while *matter* focuses on studying chemical reaction mechanisms and kinetic processes. This demands a high level of chemical expertise from the model

itself. However, the Llama3-7b model, compared to GPT-4 and Llama3-70b, has a lower reservoir of chemical knowledge, thus resulting in average performance on *atkins* and *matter*. On the other hand, the challenges in the *quan* and *chemmc* datasets lie in complex computations, prediction of molecular structures, and chemical bond reactions, emphasizing the logical reasoning ability of large models. The Llama3 experiment demonstrates that the ChemAgent framework significantly enhances the model's capability in handling logically complex chemical problems, highlighting the requirement for a certain level of subject matter expertise within the framework itself.

However, the significantly lower performance on the *atkins* dataset by Llama3-7b requires further explanation. The development set in *atkins* is much more challenging than other datasets. The solutions tend to require more reasoning steps in their decomposition, and the questions in the development set focus on a much narrower sub-topic than those in the test set. This results in a higher rate of mistakes during the development stage, leading to an initially lower-quality static memory pool. Consequently, this greatly influences and even misleads the agent's performance during the test stage.

In general, it can be seen that our approach generalizes well across different LLMs, showing consistent improvement in performance. The results indicate that our self-updating memory mechanism enhances the problem-solving capabilities of LLM-powered agents regardless of the backbone used. Notably, ChemAgent performs significantly better on GPT-4 compared to other models, suggesting that our framework becomes even more effective as base LLM models become more powerful.

Table 5: Experimental results on Llama 3.1-70b, Llama 3.1-70b and Qwen 2.5-72b. From the average improvement across four datasets, the stronger the model, the greater the improvement. Compared to the Llama 3.1-7b model, the overall enhancement using Llama 3.1-70b is better, particularly noticeable in the CHEMMC and ATKINS datasets.

| Models | CHEMMC | MATTER | ATKINS | QUAN | Avg. |
|---|---|---|---|---|---|
| *Llama 3.1-7b* | | | | | |
| Direct reasoning | 28.20 | 10.20 | **22.43** | 8.82 | 17.41 |
| ChemAgent | **56.41** | **12.24** | 19.63 | **14.71** | **25.75** |
| *Llama 3.1-70b* | | | | | |
| Direct reasoning | 33.33 | 30.61 | 30.84 | 23.53 | 29.48 |
| ChemAgent | **64.10** | **32.65** | **43.93** | **29.41** | **42.52** |
| *Qwen 2.5-72b* | | | | | |
| Direct reasoning | 48.72 | 32.65 | **57.01** | 35.50 | 43.47 |
| ChemAgent | **69.23** | **44.90** | 56.07 | **44.12** | **53.58** |

## C  PERFORMANCE BOOST: WHERE AND WHY OUR METHOD IMPROVES

### C.1  CALCULATION AND UNIT CONVERSION ARE MORE PRECISE

ChemAgent achieves notably higher accuracy in calculations and unit conversions. Two key factors contribute to this: (1) Python code is demonstrated alongside each corresponding sub-task in memory; (2) During development, ChemAgent adopted a strategy to save unit conversion steps in the long-term plan memory pool. This allows the agent to reference correct conversion steps when necessary, ensuring accurate unit conversion.

### C.2  HIGHER MEMORY'S SIMILARITY HELPS THE SOLUTION

When solving a given problem $\mathcal{P}$, a series of memories $[\mathcal{U}_1, \ldots, \mathcal{U}_n]$ may be invoked during the process. Let their similarity to the problem be represented as $[\mathcal{S}_1, \ldots, \mathcal{S}_n]$, and the mean similarity value is denoted as $\mathcal{S}_{\text{mean}, \mathcal{P}}$. In Figure 8, we visualize the distribution of $\mathcal{S}_{\text{mean}, \mathcal{P}}$, categorized by whether $\mathcal{P}$ is correctly solved. It is evident that problems with higher $\mathcal{S}_{\text{mean}, \mathcal{P}}$ are more likely to be solved correctly.

We also conducted a Chi-Square Test of Independence to assess the relationship between a similarity threshold (i.e., whether the similarity is greater than 0.805) and the correctness of the solution.

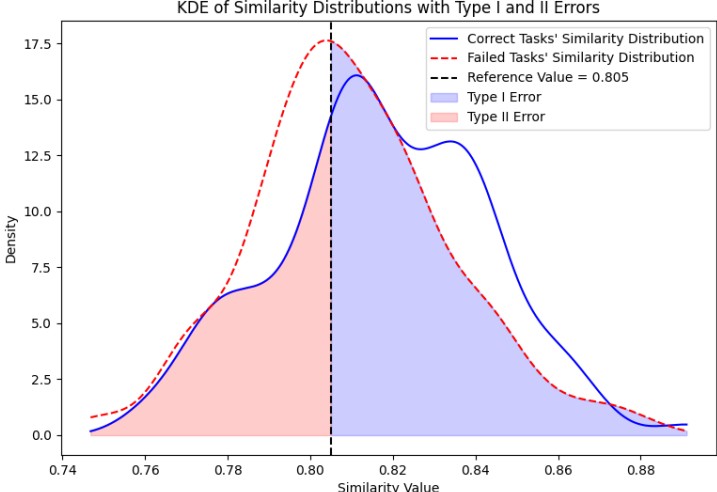

Figure 8: **Memory Similarity Analysis.** The probability density functions of invoked memory similarity for solved and failed tasks are visualized using Kernel Density Estimation. We use a reference value of 0.805 to do the Mann-Whitney U Test and get a p-value of 0.008. The similarity values are generally higher in the trajectories of solved tasks.

The Chi-Square statistic is 8.77 with a p-value of 0.003, indicating a statistically significant relationship. Thus, future work could focus on improving the similarity between invoked memories and the problem at hand to further enhance problem-solving performance.

## D  THE INFLUENCE OF THE NUMBER OF MEMORY INVOKED

We analyze the impact of the number of memory instances used on our agent's performance. In this analysis, we preserve the evaluation and refinement module since they also utilize memory. Given that the quantities of $\mathcal{M}_p$ and $\mathcal{M}_k$ are not fixed, we focus on examining how the number of $\mathcal{M}_e$ impacts performance. The results are presented in Figure 9.

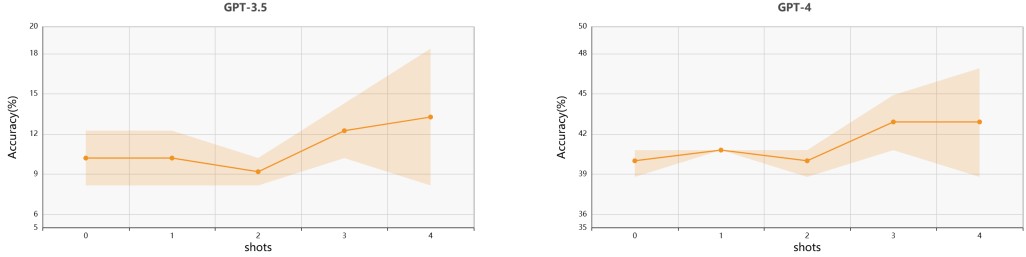

Figure 9: Test on MATTER dataset. "shots" represents the number of memory tracks given to the LLM. It demonstrates that as the number of demonstrations increases, the agent's average performance improves, but this comes at the expense of its stability.

As the number of invoked memories increases, the average accuracy improves, but the variance also grows. This indicates that although the agent's performance benefits from more memory, it also becomes more unstable. The increase in accuracy suggests that with more memory, ChemAgent adheres more strictly to the required format and acquires more useful knowledge. However, the increase in variance indicates that as the number of memory instances used increases, so too does the potential for encountering misleading or irrelevant information. Our case studies in the 4-shot

setting reveal that many memory instances carry some degree of irrelevant or *trash* information, which accumulates and can harm the agent's abilities.

Interestingly, some seemingly irrelevant information in $\mathcal{M}_e$ may enhance the creativity of the LLM when solving specific sub-tasks, resulting in a higher maximum accuracy in the 4-shot setting. This suggests that a certain level of seemingly irrelevant information can be beneficial, allowing the agent to explore hard and unknown tasks better.

# E   LIMITATION

Despite our advances, limitations include the computational intensity and time required for self-exploration, as well as the need for further refinement of memory mechanisms for tackling more complex problems. Future research should focus on understanding the specific mechanisms by which memory formation benefits reasoning, exploring how different types of memory contribute to problem-solving, and identifying optimal strategies for memory utilization. Additionally, due to limitations in budget and computational resources, we demonstrate our approach solely in the context of chemistry and have not conducted comprehensive research across the entire scientific domain, such as considering mathematics or physics.

However, we believe our proposed method has strong potential for generalization across scientific fields. We have made our code open-source and encourage future researchers to apply it to their own datasets. Specifically, adapting our method to a new domain (e.g., the CLASS sub-dataset in SciBench, related to Physics) involves modifying a small portion of the prompts (such as replacing chemistry-related sentences, e.g., "You are a Chemistry expert") and then re-starting the Library Construction phase using a development set from the new domain.

## F TASK DOMAINS OF THE DATASETS

The Chemistry domain of SciBench has four datasets, each hand-collected from four college chemistry textbooks. *Quantum chemistry (quan)* (Hair Jr et al., 2010) provides an exploration of equilibrium, structure, and reactions. *Chemistry kinetics (matter)* (Atkins et al., 2014) combines physics and mathematics, spanning through quantum mechanics and atomic structure. *Quantum mechanics (chemmc)* (McQuarrie, 2008) covers quantum mechanics and the applications in chemical bonding. *Physical chemistry (atkins)* (Atkins et al., 2023) provides explorations of equilibrium, structure, and reactions. We leverage GPT-4 to annotate each data sample in these datasets for the specific subfields. Statistically, the four datasets cover 15 chemical subfields, and the division of chemical subfields also helps us make associations in related fields to enrich the memory pool. The subfields of the datasets are shown in the figure 10.

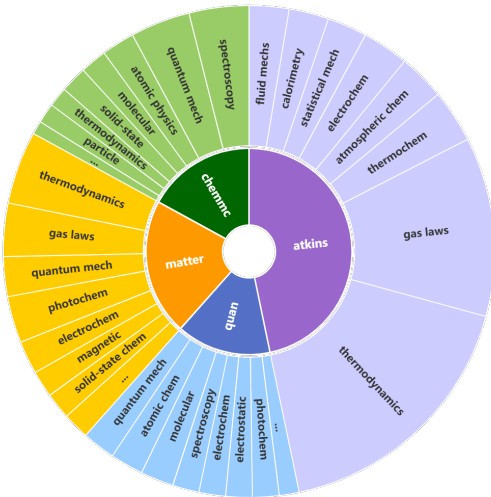

Figure 10: Chemical subfields covered by the four datasets. The subfields involved in each dataset are different, and a task may require knowledge from multiple subfields.

Table 6: Detailed statistics and information of the four datasets we experiment with. $\mathcal{D}_t$ and $\mathcal{D}_d$ refer to the number of data samples with and without solutions (development set and test set). "$\mathcal{R}_{d/t}$" means the ratio of the development set to the test set.

| Datasets | Topics | $\mathcal{D}_t$ ($\mathcal{D}_d$) | $\mathcal{R}_{d/t}$ |
|---|---|---|---|
| CHEMMC | Quantum mechanics | 39 (9) | 0.231 |
| MATTER | Chemistry kinetics | 49 (10) | 0.204 |
| ATKINS | Physical chemistry | 107 (16) | 0.150 |
| QUAN | Quantum chemistry | 34 (8) | 0.235 |

The problems in this dataset are challenging, with an average of 2 formulas and 5 steps of reasoning required to solve the problems in the experiment (Ouyang et al., 2024). Meanwhile, each dataset provides detailed step-by-step solutions for example problems, which fit well in our framework for the initial construction of memory pools, shown in Table 6.

# G PROMPTS

Here is the list of prompts we used in our study.

## G.1 INSTRUCTIONS FOR BUILDING MEMORY POOLS.

(1) Split prompts direct ChatGPT to decompose a given task into condition and problem parts. Reflect prompts are used to double-check the results of the initial decomposition to confirm that the decomposition is complete and correct and to return corrected results in the event of errors. See Figure 11.

---

**Task Split Prompt for Memory Pool Development**

**SPLIT_PROMPT_SYS**=
You are ChatGPT, a large language model trained by OpenAI, also an excellent prompt evaluator, who is capable of splitting tasks and evaluating the difficulty of tasks which will be solved by other autonomous agents powered by LLM.

**SPLIT_PROMPT** =
Now, I have a target task and its solution here: task: task solution: solution
And I also have a Python environment where agents can write code. (It is also okay not to use them and just use your knowledge):
Then, split or separate the sentence of the original task into two parts: one is conditions, and the other is questions.
Below is the format you should follow when you give the answer:

\*\*\*
CONDITIONS:
QUESTIONS:
\*\*\*

**REFLECT_PROMPT_SYS** =
You are ChatGPT, a large language model trained by OpenAI, and also an excellent answer checker who is capable of figuring out whether some conditions and questions separated from the original task are complete and correct.

**REFLECT_PROMPT** =
Now, I have a task and its corresponding conditions and questions listed below. I need your help to check whether the conditions and questions are correct and exactly fit the original task.

{{tasks}}
{{conditions}}
{{questions}}

You should give me back the conditions and questions refined according to the original task; if you think there is nothing that needs to be changed, just output the original conditions and questions.
You do NOT need to provide solutions.
All the specified numerical data should be included in the conditions part instead of the question part.
Also, the questions should NOT contain something that is not given in the condition.
Your answer should strictly follow the format below.

\*\*\*
I think the given conditions and questions fit the original task well / not well. Because ...
REFINED_CONDITIONS:

---

REFINED_QUESTIONS:
***

Figure 11: A prompt for task splitting. This prompt is mainly used to initialize the construction of the memory pool by dividing the answers and solutions given in the sample into multiple sub-problems and corresponding solutions.

(2) As a collection of problems and solutions already exists in the dataset ($\mathcal{D}_d$), we can decompose a problem and its solution at the same time. Subtask prompts decompose a complex problem into multiple sub-problems, each of which must be clearly described. The Sub_Solution prompt directs ChatGPT to decompose the solution into sub-steps corresponding to each of the sub-problems based on the set of sub-problems obtained in the previous step. Sort prompts sort the obtained subproblems from easy to difficult and provide a rationale for the sorting. See Figure 12

**Task Difficulty Ranking Prompt for Memory Pool Development**

**SUBTASK_PROMPT_SYS** =
You are an excellent expert who thinks step by step and splits a complex question into several steps.

**SUBTASK_PROMPT** =
Given a task's background and its condition: condition
Now the question is: question
And here is the given solution to this question: solution
Please think generally and step by step to divide the question into N subtasks according to the given solution. You must give each task a clear description and every subtask must be designed as a necessary step to the final question. You should strictly follow the format below:
***
I think the best way to solve the question and my solution is: ...
So I set up subtasks as described below.
TASKS=>
<st> SUBTASK 1: ... <ed>
<st> SUBTASK 2: ... <ed>
...
<st> SUBTASK N: ... <ed>
***

**SUB_SOL_PROMPT_SYS** =
You are an excellent expert who can think step by step and split a given complex solution of a certain question into several sub-steps according to the given subtasks split from the original question.

**SUB_SOL_PROMPT** =
Given a task's background and its condition:
{{condition}}
Now the question is:
{{question}}
And I have already split the question into the subtasks listed below:
{{subtask}}
And the whole solution to the original whole question is:
{{solution}}
Now, Please think generally and step by step to divide the solution into sub-solutions, which must correspond one-to-one with subtasks. You must give each sub-solution a clear description and every sub-solution should be specific and exactly solve the corresponding subtask.

For example, if there are N subtasks, you must generate N sub-solutions. And sub-solution 1 should exactly solve subtask 1.
You must generate sub-solutions according to the whole solution given. If the whole solution is "Not provided", just generate your own suggested solution for each subtask. Else if the whole solution is provided, your every sub-solution should be a part of the whole solution. You should strictly follow the format below:
***
I think the relations between subtasks and the whole solution are: ...
So, I set up sub-solutions as described below. SOLUTIONS=> <st> SUBSOLUTION 1: ... <ed> <st> SUBSOLUTION 2: ... <ed> <st> SUBSOLUTION N: ... <ed>

***

**SORT_PROMPT_SYS** =
You are ChatGPT, a large language model trained by OpenAI, and also an excellent prompt difficulty evaluator who is capable of sorting a series of tasks according to their difficulty.

**SORT_PROMPT** =
Now, I have some tasks (each containing conditions, questions, and corresponding solutions) listed below. I need your help to sort these tasks from the easiest one to the most difficult one for an agent to solve, which means you need to put the simpler subtasks in front.
{{tasks}}
Think carefully and tell me the reason why you think one is easier and another is harder.
You do not take the temporal relationship into consideration when sorting.
You do NOT need to refine the given solutions.
You do NOT need to follow the original order of the tasks.
You are FORBIDDEN to change the description of the tasks given. Just change their order.
Your answer should be a permutation of the n tasks. And the id of easier tasks should be placed in front of the harder ones. For example, if there are 3 tasks, and task 2 is the simplest, task 3 is the hardest.
Then your final output should be "RESULT=> <st> 2 1 3 <ed> "

Please strictly follow the format below.
***
I think the difficult relationship between the tasks is: ...
SO MY ANSWER IS:
RESULT=>
<st> [a permutation] <ed>
***

If there is just 1 subtask, just list it and follow this format:
***
SO MY ANSWER IS:
RESULT=>
<st> 1 <ed>
***

Figure 12: A prompt for ranking task difficulty. The subproblems are sorted in order of difficulty and solved one by one, from easy to hard, by solving the easier subproblems and learning from the experience to solve the harder ones better. This is the idea of using course learning to help agent self-evolution.

### G.2 INSTRUCTIONS FOR SOLVING PROBLEMS.

(1) Decomposition prompts (Figure 13) are used to decompose a given chemistry problem into 1-3 subtasks, each with specific goals, criticism, and milestones.

---

**Task Decomposition Prompt**

==SYSTEM_PROMPT== =
You are a Chemistry expert and an efficient plan-generation agent.
Now, you are doing an exam; you must decompose a problem into several subtasks that describe what the goals for the problem.
— Background Information —
PLAN AND SUBTASK:
A plan has a tree manner of subtasks: task 1 contains subtasks task 1.1, task 1.2, task 1.3, ... and task 1.2 contains subtasks 1.2.1, 1.2.2, ...
A subtask structure has the following JSON component:
{
"subtask name": string, name of the subtask
"goal.goal": string, the main purpose of the subtask, and what will you do to reach this goal?
"goal.criticism": string, what potential problems may the current subtask and goal have?
"milestones": list[string], what milestones should be achieved to ensure the subtask is done? And What formulas might the current subtask use? Make it detailed and specific.
}

SUBTASK HANDLE:
A task-handling agent will handle all the subtasks as the inorder-traversal. For example:
1. it will handle subtask 1 first.
2. if solved, handle subtask 2. If failed, split subtask 1 as subtask 1.1 1.2 1.3... Then handle subtask 1.1 1.2 1.3...
3. Handle subtasks recursively until all subtasks are solved. Do not make the task queue too complex, make it efficiently solve the original task.

RESOURCES:
A task-handling agent can write and execute Python code.

— Task Description —
Generate the plan for query with operation SUBTASK_SPLIT, and make sure all must-reach goals are included in the plan.

— Important Notice —
- Always make feasible and efficient plans that can lead to successful task-solving. Never create new subtasks that are similar or the same as the existing subtasks.
- For subtasks with similar goals, try to do them together in one subtask with a list of subgoals rather than split them into multiple subtasks.
- Do not waste time on making irrelevant or unnecessary plans.
- The task handler is powered by sota LLM, which can directly answer many questions. So make sure your plan can fully utilize its ability and reduce the complexity of the subtasks tree.
- You can plan multiple subtasks if you want.
- Minimize the number of subtasks, but make sure all must-reach goals are included in the plan.
- Don't generate tasks that are aimed at understanding a concept, such as "understanding the problem", the LLM who answers the task already knows the underlying concept. Check the generated subtask objectives and milestones for understanding, and regenerate the subtasks if so.
- After the subtask is generated, check to see if the answer for the task has been given in the task's known conditions. If the task has already been resolved, delete the subtask.

==USER_PROMPT== =
This is the first time you are handling the task (query), so you should give an initial plan.

---

{{similar_task_and_plan}}
Now try to use SUBTASK_SPLIT to split the following problem, here is the query which you should give an initial plan to solve:

—Your task—
{{query}}

You will use operation SUBTASK_SPLIT to split the query into 1-3 subtasks and then commit.

Figure 13: A prompt for task decomposition. Splitting the problem reduces the difficulty of solving each sub-problem. Atomized simple tasks make it easier to find trajectories of experience that can be drawn upon in the memory pool.

(2) The execution instructions (Figure 14) direct ChatGPT to solve the problem based on the current subtask and known conditions, standardize the return format that must contain the formulae used to solve the problem, a step-by-step reasoning process, and finally give a piece of Python code to arrive at the answer to the problem.

---

**Prompt for Task Execution**

**SYSTEM_PROMPT** =
A question is divided into many steps, and you will complete one of them. Please provide a clear and step-by-step solution for a scientific problem in the categories of Chemistry, Physics, or Mathematics. The problem will specify the unit of measurement, which should be included in the answer.

You have been solving a complex task by following a given plan listed below.
— Plan Overview —
The complex task has already been split into a tree-based plan as follows:
{{all_plan}}
You have already performed some of the subtasks.

**USER_PROMPT** =
Now, you continue to complete subtasks. Please combine the results of previous tasks to complete the current goal of processing subtasks. Please complete all milestones and give a concise answer that can be efficiently used by subsequent subtasks.
— Status —
Current Subtask: {{subtask_id}}
The query: {{subtask_goal}}
Milestones: {{milestones}}

Please respond strictly to the format provided. For each instance, you need to do three things.
Firstly, for "formulae retrieval", you need to identify the formulae explicitly and implicitly entailed in the problem context.
Then there is a "reasoning/calculation process" where you are required to reason step by step based on the identified formulae and problem context.
Finally, conclude the answer by writing a piece of corresponding Python code; you MUST use the International System of Units in this stage.
For each problem, the output format should incorporate the following components in the corresponding format:
**Formulae retrieval: **
[Formula 1] (the formula required to solve the problem)

[Formula 2] (the second formula required to solve the problem, if any)
...
[Formula n] (the n-th formula required to solve the problem, if any)
**Reasoning/calculation process:**
[step 1] (the first step for solving this problem)
.....
[step n] (the n-th step for solving the problem, if any)
**Answer conclusion:**
[answer] (Python code that can be executed independently)

Your answer should be a piece of Python code that solves the current question.
You must end your code by printing all the result and their units.
Make sure the code can be successfully run without any input.
Be precise. The answer should be accurate, choose the appropriate units, and prohibit the use of round functions to round off and lose the results.
Make sure the code can be successfully run without any input. And import all the modules that you need to use.

for example, you could respond to the Answer conclusion part like this:
**Answer conclusion:**
[answer]:

```
import numpy as np

# Value of 2 pi c omega_obs
omega_obs = 1.8133708490380042e+23  # Hz

# Value of D for H35Cl
D = 440.2  # kJ/mol

# Calculate beta
beta = omega_obs * (2 * D)**0.5

# Print the result
print("The value of beta is:", beta, "cm^(-1)")
```

— Similar tasks —
The following are the trajectories of tasks that have been dealt with in the past that are similar to this goal, including the successful tasks and their action lists. You can learn from them and use relevant knowledge in their procedure.
{{success_prompt}}

Figure 14: A prompt for executing tasks. The format of the specified output consists of formulas and reasoning processes. The relevant knowledge in Knowledge Memory improves the accuracy of the formula part, and the reasoning processes make the output Python code more logical.

(3) If the similarity between the retrieved memory trace and the current problem is too low, the Imagination prompts (Figure 15) direct ChatGPT to generate a problem and a solution in the specified format based on the current topic.

**Prompt for Similar Task Association**

**IMG_PROMPT** =
Please create {{top_k}} advanced chemistry questions suitable for in-depth understanding and application of chemical formulas and principles. Each question should focus on the deeper aspects of the [topic] provided to understand the principles of chemistry and the

reasoning process.
[topic]: {{topic}}
Guidelines for Problem Creation:
- Use of Samples: You are provided with sample questions for reference. Feel free to use these to guide the style and depth of the problems.
- Beyond the Examples: You are encouraged to use your background and expertise to create problems that go beyond the provided examples, ensuring the problems are as diverse and comprehensive as possible.

Requirements for Each Problem:
a. Problem Statement: Clearly define the challenge or task.
b. Solution: Provide a detailed solution that includes:
I. Formulas and Knowledge Needed: List the equations and concepts required to understand and solve the problem.
II. Reasoning Steps: Outline the logical or mathematical steps to solve the problem.
III. Python code: Executable Python code is generated to solve problems. At the end of each problem, please include Python code that can be used to confirm and verify the correctness of the provided solution. The Python solutions should illustrate the entire solution process, from the initial step to the final answer, rather than merely validating the result. Develop these solutions such that each step of the mathematical process is explicitly demonstrated and calculated in Python. Additionally, ensure that you run your Python code to confirm it is free from any errors.
d. Diversity: Ensure a wide range of problems, each focusing on different elements from the subtopic list.
e. Presentation: Please output your problem statement, solution, detailed explanation, and a self-contained Python code for verification below in the specified format.

For each generated question, the output is required to be in the following format:
[Task Start]
[Problem Statement]: (your problem)

**Formulae retrieval: **
[Formula 1] (the formula required to solve the problem)
[Formula 2] (the second formula required to solve the problem, if any) ...
[Formula n] (the n-th formula required to solve the problem, if any)

**Reasoning/calculation process:**
[step 1] (the first step for solving this problem)
.....
[step n] (the n-th step for solving the problem, if any)

**Answer conclusion:**
[answer]: (Python code that can be executed independently)
[Task End]

You have to generate {{topk}} tasks about {{topic}}.
Sample demonstration example:
{{example_shots}}

Figure 15: A prompt for associating similar tasks. The dataset covered 15 sub-domains, and the question was modeled by interrogation to determine which sub-domain the question belonged to. The associatively generated trajectories must be in the same format as the trajectories in Execution Memory.

(4) The evaluation prompt (Figure 16) is used to assess the correctness of the current solution, including judging the correctness of the given formulas, the rigor of the reasoning process, and whether the output of the Python code meets the task goals and completes all task milestones.

---

**Prompt for Task Evaluation**

**SCORE_PROMPT** =
You tackled a sub-problem in a chemistry problem; the format of the solution to the problem is **Formulae retrieval: ** and **Reasoning/calculation process:** and **Answer conclusion:**(which includes a piece of Python code and its corresponding output).

—Subtask—
The question: {subtask_goal}
Milestones: {milestones}

[Response Start]
{response}
[Response End]

For each instance, you need to do four things:
- First, judge whether the given formula is correct and whether the constants are correct.
- Second, judge whether the reasoning process is rigorous and correct.
-Third, determine whether the Python function outputs the parameters required by the task goal and milestone.
- Finally, score the degree of completion and correctness of the whole task. You should give the "confidence score" on the scale of [0,1]. Please be very strict about your ratings.

The output format should incorporate these components in the following format:
**Judgement of the retrieved formulae:**
[judgement] (Your assessment of whether the retrieved formulae are correct or not.)

**Judgement of the reasoning process:**
[judgement] (Your assessment of whether the reasoning process is correct or not.)

**Judgement of the Answer conclusion:**
[judgement] (Your assessment of whether the Python code outputs the parameters required in the task objective and whether the Python code correctly infers according to the analysis in reason.)

**Confidence score:**
[score] (float number in [0,1], A very strict score is given to the correctness of the solution of the entire task)

---

Figure 16: A prompt for evaluating tasks. Evaluate the current answer, generate answers consecutively, and select the answer with the highest score.

(5) Finally, the final answer and summary of posterior knowledge are obtained by summary prompts (Figure 17). Summarize the total process of solving the problem, the relevant knowledge used, and the formulas that can be used to enrich the memory pool. At the same time, the answers to all sub-problems are combined to produce the final answer to the whole problem.

---

**Prompt for Task Summary**

**SYSTEM_PROMPT** =
You are a posterior_knowledge_obtainer.
Now that you've completed a parent task, the task is made up of many subtasks, each of which has been completed.

Your plan for the parent task is as follows:
—Parent Goal —
{{parent_goal}}
— Sub-task division —
{{all_plan}}

The flow of your actions for handling subtasks is:
— Action lists for Subtasks —
{{sub_plan}}

Now, you have to learn some posterior knowledge from this process by doing the following things:
1. Summary: Summarize the ideas of all subtasks to the parent task, and summarize a total process to the parent task according to the action process of each subtask. Explicitly include all the formulas used during performing the subtasks in this summary. Specific numbers and numerical results are NOT needed in this part.

2. Reflection of knowledge: After performing this task, you get some knowledge of generating plans for similar tasks. Only conclude the used knowledge and formulas used in this whole task, do NOT contain numerical calculation process and results.

3. Final Answer: Give the final answer to the task according to the course of the task, and ask the answer to be very short, without explaining the reason and adding unnecessary punctuation. If it's a math problem, only the last value is given.

---

Figure 17: A prompt for summarizing tasks. Summarizing tasks is crucial for the self-evolution of ChemAgent. Summarize the formulas and principles of this task and add them to the Memory Library.

(6) When a task fails to execute, it may be because the task decomposition is unreasonable. The framework can perform adjustment operations such as adding/splitting/deleting the current task tree to adjust the overall task framework. See Figure 18 for the exact prompt.

---

**Prompt for Task Refinement**

**SYSTEM_PROMPT** =
You are a plan-rectify agent, your task is to iteratively rectify a plan of a query.
— Background Information —
PLAN AND SUBTASK:
A plan has a tree manner of subtasks: task 1 contains subtasks task 1.1, task 1.2, task 1.3, ... and task 1.2 contains subtasks 1.2.1, 1.2.2, ...
A subtask structure has the following JSON component:
{
"subtask name": string, name of the subtask
"goal.goal": string, the main purpose of the subtask, and what will you do to reach this goal?
"goal.criticism": string, what potential problems may the current subtask and goal have?

---

"milestones": list[string], what milestones should be achieved to ensure the subtask is done? And What formulas might the current subtask use? Make it detailed and specific.
}

SUBTASK HANDLE:
A task-handling agent will handle all the subtasks as the inorder-traversal. For example:
1. it will handle subtask 1 first.
2. if solved, handle subtask 2. If failed, split subtask 1 as subtask 1.1 1.2 1.3... Then handle subtask 1.1 1.2 1.3...
3. Handle subtasks recursively until all subtasks are solved. Do not make the task queue too complex, make it efficiently solve the original task.

RESOURCES:
A task-handling agent can write and execute Python code.

— Task Description —
Your task is iteratively to rectify a given plan based on the goals, suggestions, and now handling positions.

In this mode, you will use the given operations to rectify the plan. At each time, use one operation.
SUBTASK OPERATION:
1. split: Split an already handled but failed subtask into subtasks because it is still so hard. The "target_subtask_id" for this operation must be a leaf task node that has no children subtasks, and should provide new split "subtasks" of length 2-4. You must ensure the "target_subtask_id" exists, and the depth of new split subtasks < {{max_plan_tree_depth}}.
    - split 1.2 with 2 subtasks will result in create new 1.2.1, 1.2.2 subtasks.

2. add: Add new subtasks as brother nodes of the 'target_subtask_id'. This operation will expand the width of the plan tree. The 'target_subtask_id' should point to a now-handling subtask or future subtask.
    - add 1.1 with two subtasks will result in creating new 1.2, 1.3 subtasks.
    - add 1.2.1 with 3 subtasks wil result in create new 1.2.2, 1.2.3, 1.2.4 subtasks.

3. delete: Delete a subtask. The 'target_subtask_id' should point to a future/TODO subtask. Don't delete the now handling or done subtask.
    - delete 1.2.1 will result in delete 1.2.1 subtask.

— Note —
The user is busy, so make efficient plans that can lead to successful task-solving.
Do not waste time making irrelevant or unnecessary plans. Don't use search engines since you know about planning. Don't divide trivial tasks into multiple steps.
If the task is unsolvable, give up and submit the tas k.

*** Important Notice ***
- Never change the subtasks before the handling positions, you can compare them in lexicographical order.
- Never create (with add or split action) new subtasks that are similar or the same as the existing subtasks.
- For subtasks with similar goals, try to do them together in one subtask with a list of subgoals rather than split them into multiple subtasks.
- Every time you use an operation, make sure the hierarchy structure of the subtasks remains, e.g., if subtask 1.2 is to "find A,B,C" , then the newly added plan directly related to this plan (like "find A", "find B", "find C") should always be added as 1.2.1, 1.2.2, 1.2.3...
- You are restricted to give operations in at most 4 times, so the plan refinement is not so much.
- The task handler is powered by sota LLM, which can directly answer many questions. So

make sure your plan can fully utilize its ability and reduce the complexity of the subtasks tree.

**USER_PROMPT=**
Your task is to choose one of the operators of SUBTASK OPERATION, note that
1.You can only modify the subtask with subtask_id > {{subtask_id}}(not included).
2. Please use a function call to respond to me (remember this!!!).

Figure 18: A prompt for refining tasks. Incorrect answers may result from an error in a sub-question, leading to subsequent inaccuracies, or from a misjudgment in the decomposition of the question. In the former case, the Task Evaluation module is used to enhance the accuracy of problem-solving, while the Task Refinement module is employed to modify the question decomposition tree in the latter case.

### G.3 INSTRUCTIONS FOR BASELINE

Both direct-reasoning and fewshot+Python setups are used as baselines in the main experiment, mainly referring to the experimental setup in SciBench Wang et al. (2024a). The following instruction (Figure 19) is used in the fewshot+Python setup, which asks for a piece of Python code to be output and is followed by three demonstrations.

---

**Prompt for Few-Shot Python**

**SYS_PROMPT=**
Please provide a clear and step-by-step solution for a scientific problem in the categories of Chemistry, Physics, or Mathematics. The problem will specify the unit of measurement. For clarity, please translate the solution steps into Python code and encase the Python code within triple backticks.

---

Figure 19: A prompt for Few-shot+Python. The baseline setup directly requires LLMs to generate Python code, and through few-shot learning, the responses are standardized. This approach, compared to the direct-reasoning setup, has shown an improvement in correctness.

## H  SPECIFIC CASE TRAJECTORIES

To concretely demonstrate the problem-solving process of our framework, we provide a complete execution trajectory of a successfully answered question to illustrate the procedural workflow of our framework.

---

**Task Description**

At $300\,\text{K}$ and $20\,\text{atm}$, the compression factor of a gas is 0.86. Calculate the volume occupied by $8.2\,\text{mmol}$ of the gas under these conditions. The unit of the answer should be $\text{cm}^3$.

**Task Decomposition**

**Task 1: Calculate the volume in liters**

Use the definition of the compression factor and the ideal gas law to calculate the volume occupied by $8.2\,\text{mmol}$ of the gas under the given conditions in liters.

**Task 2: Convert the volume to $\text{cm}^3$**

Convert the calculated volume from liters to cubic centimeters.

**Execution of Task 1 - Calculate the Volume in Liters**

**Formulas:**
- Ideal Gas Law: $PV = nRT$
- Compression Factor: $Z = \dfrac{PV}{nRT}$

**Given:**
- Compression factor, $Z = 0.86$
- Pressure, $P = 20\,\text{atm}$ (convert to Pa)
- Number of moles, $n = 8.2\,\text{mmol} = 8.2 \times 10^{-3}\,\text{mol}$
- Gas constant, $R = 8.3145\,\text{J}\,\text{K}^{-1}\,\text{mol}^{-1}$
- Temperature, $T = 300\,\text{K}$

**Calculations:**
1. Convert pressure from atm to Pa:
   $P = 20\,\text{atm} \times 101325\,\text{Pa/atm} = 2,026,500\,\text{Pa}$
2. Rearrange the compression factor formula to solve for $V$:
   $Z = \dfrac{PV}{nRT} \implies V = \dfrac{ZnRT}{P}$
3. Calculate the volume in cubic meters:
   $V = \dfrac{0.86 \times 8.2 \times 10^{-3}\,\text{mol} \times 8.3145\,\text{J}\,\text{K}^{-1}\,\text{mol}^{-1} \times 300\,\text{K}}{2,026,500\,\text{Pa}}$
4. Compute $V$:
   $V = \dfrac{0.86 \times 8.2 \times 10^{-3} \times 8.3145 \times 300}{2,026,500}$
   $V \approx 8.6801 \times 10^{-6}\,\text{m}^3$
5. Convert volume to liters (since $1\,\text{m}^3 = 1000\,\text{L}$):
   $V = 8.6801 \times 10^{-6}\,\text{m}^3 \times 1000\,\text{L/m}^3 = 0.0086801\,\text{L}$

**Execution of Task 2 - Convert the Volume to $\text{cm}^3$**

**Formula:**
- Conversion from liters to cubic centimeters: $1\,\text{L} = 1000\,\text{cm}^3$

**Calculation:**
1. Convert volume from liters to cubic centimeters:
   $V = 0.0086801\,\text{L} \times 1000\,\text{cm}^3/\text{L} = 8.6801\,\text{cm}^3$

**Summary Results**

The volume occupied by $8.2\,\text{mmol}$ of the gas under the given conditions is approximately **$8.6801\,\text{cm}^3$**.

---

We also provide an execution trajectory of a successfully answered question in another benchmark, ChemBench (Mirza et al., 2024) which direct-reasoning method fails to solve. The backbone LLM used here is gpt-4o (2024-08-01-preview).

---

**Task Description**

---

Statement 1: Addition of a non-reacting gas at constant volume and constant temperature to a container containing gas (X) does not change the partial pressure of gas (X). Statement 2: The addition of a non-reacting gas to a container at constant volume and constant temperature results in an increase in the total pressure of the container, but not in the partial pressure of the gases in the container. Which of the following statements is the most correct option for the given two statements?

A.Statement 1 is False, Statement 2 is True.

B.Statement 1 is True, Statement 2 is False.

C.Statements 1 and 2 are True. Statement 2 is NOT the correct explanation for Statement 1.

D.Statements 1 and 2 are True. Statement 2 is the correct explanation for Statement 1.

Choose one correct choice.

## Task Decomposition

### Task 1: Verify the validity of Statement 1

Determine effect of adding a non-reacting gas on partial pressure of gas (X) at constant volume and temperature.

### Task 2: Verify the validity of Statement 2

Determine effect of adding a non-reacting gas on total pressure at constant volume and temperature.

### Task 3: Compare Statements and Choose the Correct Option

Compare the findings from the analysis of Statements 1 and 2 and determine the most correct option from the provided choices.

## Execution of Task 1 - Verify the validity of Statement 1

**Formulas:**

- Ideal Gas Law: $PV = nRT$

- $P_{\text{partial}} = \frac{n_{\text{gas}}RT}{V}$

**Given:**

- $P_{\text{partial}}$ is the partial pressure of gas (X)

- $n_{\text{gas}}$ is the number of moles of gas (X)

- $T$ is the temperature, $V$ is the volume

**Calculations:**

1. The partial pressure of a gas in a container is given by the ideal gas law, which states that the partial pressure is directly proportional to the number of moles of the gas, the temperature, and inversely proportional to the volume.

$$P_{\text{partial}} = \frac{n_{\text{gas}}RT}{V}$$

2. When a non-reacting gas is added to the container at constant volume and constant temperature, the number of moles of gas (X) does not change. Therefore, the partial pressure of gas (X) remains the same because $n_{\text{gas}}$, $R$, $T$, and $V$ are constant.

The initial and final partial pressures of gas (X) are the same, verifying that Statement 1 is true.

## Execution of Task 2 - Verify the validity of Statement 2

**Formula:**

- The total pressure of a gas mixture is the sum of the partial pressures of the individual gases:

$$P_{\text{total}} = P_1 + P_2 + \ldots + P_n$$

**Calculation:**

1. Consider a container with gas (X) at constant volume and temperature. The initial total pressure is the partial pressure of gas (X), $P_X$.

2. When a non-reacting gas (Y) is added to the container at constant volume and temperature, the partial pressure of gas (X) remains the same, $P_X$.

3. The total pressure of the container after adding gas (Y) is the sum of the partial pressures of gas (X) and gas (Y), $P_{\text{total}} = P_X + P_Y$.

## Execution of Task 3 - Compare Statements and Choose the Correct Option

**Formula:**

No specific formula is needed for this subtask.

**Reasoning:**
1. Analyze the results from the previous subtasks: - From Task 1.1: The partial pressure of gas (X) remains unchanged after adding a non-reacting gas at constant volume and temperature. - From Task 1.2: The total pressure increases after adding a non-reacting gas at constant volume and temperature, but the partial pressure of gas (X) remains unchanged.
2. Compare the findings from the analysis of Statements 1 and 2.
3. Determine the most correct option based on the analysis: - Both statements are true, and Statement 2 correctly explains Statement 1.

**Summary Results**

D. Statements 1 and 2 are True. Statement 2 is the correct explanation for Statement 1.

An example result of the decomposition step of the library construction phase, which help understand how the sub-tasks are organized.

**Original Task**

The change in molar internal energy when $CaCO_3(s)$ as calcite converts to another form, aragonite, is $+0.21\text{ kJ mol}^{-1}$. Calculate the difference between the molar enthalpy and internal energy changes when the pressure is 1.0 bar given that the densities of the polymorphs are $2.71\text{ g cm}^{-3}$ and $2.93\text{ g cm}^{-3}$, respectively.

**Original Solution**

The change in enthalpy when the transition occurs is

$$\Delta H_{\mathrm{m}} = H_{\mathrm{m}}(\text{ aragonite }) - H_{\mathrm{m}}(\text{ calcite })$$
$$= \{U_{\mathrm{m}}(\mathrm{a}) + pV_{\mathrm{m}}(\mathrm{a})\} - \{U_{\mathrm{m}}(\mathrm{c}) + pV_{\mathrm{m}}(\mathrm{c})\}$$
$$= \Delta U_{\mathrm{m}} + p\{V_{\mathrm{m}}(\mathrm{a}) - V_{\mathrm{m}}(\mathrm{c})\}$$

where a denotes aragonite and c calcite. It follows by substituting $V_{\mathrm{m}} = M/\rho$ that

$$\Delta H_{\mathrm{m}} - \Delta U_{\mathrm{m}} = pM\left(\frac{1}{\rho(\mathrm{a})} - \frac{1}{\rho(\mathrm{c})}\right)$$

Substitution of the data, using $M = 100\text{ g mol}^{-1}$, gives

$$\Delta H_{\mathrm{m}} - \Delta U_{\mathrm{m}} = \left(1.0 \times 10^5\text{ Pa}\right) \times \left(100\text{ g mol}^{-1}\right) \times \left(\frac{1}{2.93\text{ g cm}^{-3}} - \frac{1}{2.71\text{ g cm}^{-3}}\right)$$
$$= -2.8 \times 10^5\text{ Pa cm}^3\text{ mol}^{-1} = -0.28\text{ Pa m}^3\text{ mol}^{-1}$$

**Decomposition Results**

**SUB-TASK 1**

*Condition 1*

The change in molar internal energy when $CaCO_3(s)$ as calcite converts to another form, aragonite, is $+0.21\text{ kJ/mol}$. The pressure is 1.0 bar. The densities of the polymorphs are $2.71\text{ g/cm}^3$ and $2.93\text{ g/cm}^3$, respectively.

*Question 1*

Identify the given data and the required physical quantities (molar internal energy change, pressure, densities, etc.).

*Solution 1*

$$\rho_{\text{calcite}} = 2.71\text{ g/cm}^3, \quad \rho_{\text{aragonite}} = 2.93\text{ g/cm}^3$$

**SUB-TASK 2**

*Condition 2*

The change in molar internal energy when $CaCO_3(s)$ as calcite converts to another form, aragonite, is $+0.21\text{ kJ/mol}$. The pressure is 1.0 bar. The densities of the polymorphs are $2.71\text{ g/cm}^3$ and $2.93\text{ g/cm}^3$, respectively.

*Question 2*

Express the molar volumes of calcite and aragonite in terms of their densities and the molar mass of $CaCO_3$.

*Solution 2*

$$V_m(\text{calcite}) = \frac{M}{\rho_{\text{calcite}}}, \quad V_m(\text{aragonite}) = \frac{M}{\rho_{\text{aragonite}}}$$

where $M$ is the molar mass of $CaCO_3$ (100 g/mol).

**SUB-TASK 3**

*Condition 3*

The change in molar internal energy when $CaCO_3(\text{s})$ as calcite converts to another form, aragonite, is $+0.21 \, \text{kJ/mol}$. The pressure is 1.0 bar. The densities of the polymorphs are $2.71 \, \text{g/cm}^3$ and $2.93 \, \text{g/cm}^3$, respectively.

*Question 3*

Substitute the given densities and molar mass into the molar volume expressions to find the specific volumes.

*Solution 3*

$$V_m(\text{calcite}) = \frac{100 \, \text{g/mol}}{2.71 \, \text{g/cm}^3}, \quad V_m(\text{aragonite}) = \frac{100 \, \text{g/mol}}{2.93 \, \text{g/cm}^3}$$

**SUB-TASK 4**

*Condition 4*

The change in molar internal energy when $CaCO_3(\text{s})$ as calcite converts to another form, aragonite, is $+0.21 \, \text{kJ/mol}$. The pressure is 1.0 bar. The densities of the polymorphs are $2.71 \, \text{g/cm}^3$ and $2.93 \, \text{g/cm}^3$, respectively.

*Question 4*

Calculate the change in volume between aragonite and calcite using the specific volumes.

*Solution 4*

$$\Delta V_m = \left( \frac{100 \, \text{g/mol}}{2.93 \, \text{g/cm}^3} \right) - \left( \frac{100 \, \text{g/mol}}{2.71 \, \text{g/cm}^3} \right)$$

$$\Delta V_m \approx 34.13 \, \text{cm}^3/\text{mol} - 36.90 \, \text{cm}^3/\text{mol} = -2.77 \, \text{cm}^3/\text{mol}$$

**SUB-TASK 5**

*Condition 5*

The change in molar internal energy when $CaCO_3(\text{s})$ as calcite converts to another form, aragonite, is $+0.21 \, \text{kJ/mol}$. The pressure is 1.0 bar. The densities of the polymorphs are $2.71 \, \text{g/cm}^3$ and $2.93 \, \text{g/cm}^3$, respectively.

*Question 5*

Write down the equation relating the change in molar enthalpy to the change in molar internal energy and the pressure-volume work term.

*Solution 5*

$$\Delta H_m = \Delta U_m + p \Delta V_m$$

where $\Delta V_m = V_m(\text{aragonite}) - V_m(\text{calcite})$.

**SUB-TASK 6**

*Condition 6*

The change in molar internal energy when $CaCO_3(\text{s})$ as calcite converts to another form, aragonite, is $+0.21 \, \text{kJ/mol}$. The pressure is 1.0 bar. The densities of the polymorphs are $2.71 \, \text{g/cm}^3$ and $2.93 \, \text{g/cm}^3$, respectively.

*Question 6*

Substitute the values of pressure, molar internal energy change, and the change in volume into the enthalpy change equation to find the difference between the molar enthalpy and internal energy changes.

*Solution 6*

$$\Delta H_m - \Delta U_m = p\Delta V_m$$
$$\Delta H_m - \Delta U_m = (1.0 \times 10^5 \, \text{Pa}) \times (-2.77 \times 10^{-6} \, \text{m}^3/\text{mol})$$
$$\Delta H_m - \Delta U_m = -0.277 \, \text{kJ/mol}$$

**SUB-TASK 7**
*Condition 7*
The change in molar internal energy when $CaCO_3(s)$ as calcite converts to another form, aragonite, is $+0.21 \, \text{kJ/mol}$. The pressure is 1.0 bar. The densities of the polymorphs are $2.71 \, \text{g/cm}^3$ and $2.93 \, \text{g/cm}^3$, respectively.
*Question 7*
Verify the units of the calculated result to ensure consistency with the required units (e.g., $\text{kJ/mol}$).
*Solution 7*
The units of the result $-0.277 \, \text{kJ/mol}$ are correct and consistent with the required units.

**SUB-TASK 8**
*Condition 8*
The change in molar internal energy when $CaCO_3(s)$ as calcite converts to another form, aragonite, is $+0.21 \, \text{kJ/mol}$. The pressure is 1.0 bar. The densities of the polymorphs are $2.71 \, \text{g/cm}^3$ and $2.93 \, \text{g/cm}^3$, respectively.
*Question 8*
Summarize the final answer with the correct sign and units.
*Solution 8*
The difference between the molar enthalpy change and the molar internal energy change is

$$\Delta H_m - \Delta U_m = -0.28 \, \text{kJ/mol}.$$

