# OpenReview forum: "ChemAgent: Self-updating Memories in Large Language Models Improves Chemical Reasoning"
_ICLR.cc/2025/Conference — ICLR 2025 Poster_

### Official Review · Reviewer_h1D2 · 2024-10-30

**Soundness:** 3
**Presentation:** 2
**Contribution:** 3
**Rating:** 6
**Confidence:** 5

**Summary:**

The paper introduces ChemAgent, a method for improving LLMs reasoning in chemistry-related tasks based on the use of tsak decomposition and memory modules of three kinds.
In particular, the authors implement a system with 3 kinds fo memory, namely knowledge memory Mk, plan memory Mp, and Execution memory Me, each of which plays a role in different parts of the pipeline.
Mp is used to aid an LLM when decomposing a given task by providing it with previous plans used to solved other tasks. Mk is used to retrieve facts and other relevant things useful to tackle specific sub-tasks, and Me provides detailed solutions to similar problems.
The evaluations are done mostly over the recently published SciBench benchmark. The authors perform multiple ablations to assess the effect of each component of the pipeline including each memory module, the LLM used, the effect of accumulating memores in Mp and Me. In addition, the papers presents comparisons with other SOTA methods and show that their method outperforms previous approaches.

**Strengths:**

The method builds up on other work (particularly StructChem), by recognizing where that approach fails and facilitating different modules to support the LLM during the reasoning process. Each memory module is designed to tackle an aspect which I find interesting, in particular the knowledge memory module (Mk) tackles the issue of bad recall of facts, such as constants, formulas, etc, preventing hallucinations of this type.
Plan memory and execution memory both aid the LLM at the task of splitting tasks into relevant and actionable plans, based on previous experience.
The evaluations show the strengths of the system and an array of ablations is performed to assess the influence of each module, and of the quality of each module, in the resulting performance.
Evaluation of the effect of evolution in memory as the agent solves tasks (Section 3.3) is very interesting, and it's indeed an experiment that I was looking forward to seeing while reading the paper. It proves one of the main claims: that experience (through expansion of the memory modules) can improve the performance of the agent.
The error analysis (Section 3.5) is very interesting and gives a lot of insight into where and why the agent fails at solving tasks. This is important for further developments as it gives an intuition on what parts need fixing or improving.

**Weaknesses:**

## Notes on the field specific Agent

- there is nothing that makes the system chemistry-specific, other than the type of problems that are collected in its memory components. In fact, given the content of the paper I can imagine this system can be used for physics, maths (and thus evaluated in SciBench) or even extended to more general reasoning questions. In that sense, the relevance of this work can be improved by adding more extensive experiments on other fields outside of chemistry, as there is no chemistry-specific insight in the current formulation.

- On the other hand, given that this work focuses largely on the domain of chemistry, it would be good to see more work into chemistry-specific insights or inductive biases, inclusion of modalities (that for instance improve performance on understanding of molecules, reactions, results from analytical instruments), etc, that serve as an argument for why the focus is so strongly placed in chemistry. Again, this same system could perfectly be applied for physics or maths, however no evaluation shows this.

## Missing citations, ignoring several other works and benchmarks

- The concepts of "types of memory" is not new and, although the paper cites some relevant papers, it's missing a few that implement similar ideas on other domains, however the integration of all into this complete framework is new, and interesting approach to reasoning of LLMs.
See the references given at the bottom [1-5], along with others that I didn't include here. There's even a whole review on memory mechanisms behind LLM Agents, that haven been cited here and that I find extremely relevant.
- Omission of other benchmarks: Given that the system is designed for chemistry-specific tasks, and given the low amount of benchmarks specific for chemical reasoning, it would be good to assess ChemAgent's performance on other benchmarks, such as ChemBench [7]
- For being so chemistry-specific, this work misses to cite foundational works that are related, such as [8-10] and others. [11] is only tangentially mentioned at the appendix as a related work. The paper in general misses such kind of extremely relevant references.
- Some important citations are only tangentially relevant. Particularly in **line 048**, for the statement "previous approaches in chemical reasoning tasks have focused on decomposing the reasoning process", none of the references cited are relevant. Feinberg 2018 (PotentialNet) is a method for molecular property prediction. Yao 2024 (TreeofThoughts) and Zhang 2023 (Automatic Chain of Thought Prompting) have nothing to do with chemical reasoning. Only Zhong 2023 is directly relevant to this statement.


## Writing

Some parts of the paper could use rewriting for improving clarity, namely:

- line 044: unclear statement: llms struggle to (1) use formulas (2) exhibit incorrect steps. Should be llms (1) struggle to… (2) exhibit…
- line 052: It is mentioned that "previous methods rely on predefined templates", and cites self-consistency paper [12] as an example of that. Furthermore this is presented as an advantage of the current method over others, however: 1) ChemAgent also relies heavily on predefined prompts, and 2) self-consistency is a general methodology that considers multiple outputs from an LLM to yield the most prominent answer from multiple reasoning paths. In that sense, it's not clear what is understood by "pre-defined templates", and why this is a drawback of other methods that is somehow addressed in this work.
- In Table 1 it is very unclear where results come from, or how they have been calculated. Especially for the baselines with gpt4 (direct reasoning), these results mismatch those presented in the original paper for the benchmark. Only the results for “few-shot + python” have been taken from the original benchmakr paper. Further inspection indicates that the results reported for “Direct reasoning, GPT-4” have been directly taken from the StructChem paper [13], however no explicit mention is made of this. Additionally, these results are taken from the row “Few shot setting - Direct reasoning” in Table 1 of [13], which is different from the described “Direct reasoning”. Please cite the specific sources for each of the results in the table.


## Missing ablations

- The evaluation and refinement module uses gpt-4, regardless of the LLM used as a base. It is critical to evaluate the effect of this design choice, as it's not clear how much gpt-4 as evaluator improves gpt-3.5's (or any other llm's) performance.
-



[1] Zhang Y, Wang C, Qi J,Peng Y. Leave It to Large Language Models! Correction and Planning with Memory Integration. Cyborg Bionic Syst. 2024;5:Article 0087. https://doi.org/10.34133/cbsystems.0087
[2] Li, Kun, et al. "Vector Storage Based Long-term Memory Research on LLM" International Journal of Advanced Network, Monitoring and Controls, vol. 9, no. 3, Sciendo, 2024, pp. 69-79. https://doi.org/10.2478/ijanmc-2024-0029
[3] Zhong, W., Guo, L., Gao, Q., Ye, H., & Wang, Y. (2024, March). Memorybank: Enhancing large language models with long-term memory. In Proceedings of the AAAI Conference on Artificial Intelligence (Vol. 38, No. 17, pp. 19724-19731).
[4] Huang, Xu et al. “Understanding the planning of LLM agents: A survey.” ArXiv abs/2402.02716 (2024)
[5] Guo, J., Li, N., Qi, J., Yang, H., Li, R., Feng, Y., ... & Xu, M. (2023). Empowering Working Memory for Large Language Model Agents. arXiv preprint arXiv:2312.17259.
[6] Zhang, Zeyu et al. “A Survey on the Memory Mechanism of Large Language Model based Agents.” ArXiv abs/2404.13501 (2024)
[7] Mirza, A., Alampara, N., Kunchapu, S., Emoekabu, B., Krishnan, A., Wilhelmi, M., ... & Jablonka, K. M. (2024). Are large language models superhuman chemists?. arXiv preprint arXiv:2404.01475.
[8] Boiko, D.A., MacKnight, R., Kline, B. et al. Autonomous chemical research with large language models. Nature 624, 570–578 (2023). https://doi.org/10.1038/s41586-023-06792-0
[9] Darvish, K., Skreta, M., Zhao, Y., Yoshikawa, N., Som, S., Bogdanovic, M., ... & Shkurti, F. (2024). ORGANA: A Robotic Assistant for Automated Chemistry Experimentation and Characterization. arXiv preprint arXiv:2401.06949.
[10] Skreta, Marta et al. “RePLan: Robotic Replanning with Perception and Language Models.” ArXiv abs/2401.04157 (2024)
[11] M. Bran, A., Cox, S., Schilter, O. et al. Augmenting large language models with chemistry tools. Nat Mach Intell 6, 525–535 (2024). https://doi.org/10.1038/s42256-024-00832-8
[12] Wang, X., Wei, J., Schuurmans, D., Le, Q., Chi, E., Narang, S., ... & Zhou, D. (2022). Self-consistency improves chain of thought reasoning in language models. arXiv preprint arXiv:2203.11171.
[13] Ouyang, S., Zhang, Z., Yan, B., Liu, X., Choi, Y., Han, J., & Qin, L. (2023). Structured chemistry reasoning with large language models. arXiv preprint arXiv:2311.09656.

**Questions:**

1. Given that the system doesn't appear to have chemistry-specific components, is there any particular reason why you focused solely on chemistry applications? Have you considered or tested the system's performance in other domains like physics or mathematics?
2. Can you provide more details on how the results in Table 1 were obtained, particularly for the GPT-4 baselines? It appears some results may have been taken from other papers - could you clarify the sources and methodologies used?
3. Have you considered evaluating ChemAgent's performance on other chemistry-specific benchmarks, such as ChemBench? What was the rationale for the choice of benchmarks used in the current study?
4. Could you elaborate on what you mean by "predefined templates" when discussing previous methods? How does ChemAgent's approach differ, given that it also relies on predefined prompts?
5. Several relevant works in the field of chemical reasoning and LLM agents with memory mechanisms seem to be missing from your citations. Please properly address these works and cite them accordingly.

---

> ### Author Response · Authors · 2024-11-24
> **Response to Reviewer h1D2**
>
> We greatly appreciate your detailed feedback and insightful questions. Below, we address each concern, elaborating on improvements made and additional experiments conducted.
>
> ### **Q1. Why focus solely on chemistry applications?**
>
> Our initial focus on chemistry stems from its unique combination of challenges:
>
> 1. **Complexity**: Chemistry tasks require both rather **long reasoning chains** and the integration of domain-specific knowledge, such as formulas and constants.
>
> 2. **Reasoning vs. Calculation**: Unlike advanced mathematics, where highly complex and precise calculations (e.g., PDEs) are common, chemistry calculations are often simpler, making them ideal for validating our framework's **reasoning capabilities**.
>
> 3. **Data availability**: Chemistry benchmarks like SciBench offer rich, domain-specific evaluation datasets.
>
> ### **Q2. Is there anything chemistry-specific about the system?**
>
> The answer is nuanced, while ChemAgent's framework can smoothly extend to other domains, its **memory library** is tailored to chemistry during the **library construction phase**.
> For example:
> - **Execution Memory (M_e)** stores solutions to chemical problems.
>  - **Plan Memory (M_p)** is built based on chemistry-specific task decompositions.
> - Transitioning to a different domain (e.g., physics or mathematics) would require restarting the memory initialization process. Without this, applying chemistry-specific memories to other domains could degrade performance.
>
> ### **Q3. Performance in other domains**
>
> To address this, we tested **GPT-4omini** on other SciBench subdomains, including physics (**CLASS**) and mathematics (**DIFF** and **STAT**).
>
> Results are shown below:
>
> | Method                     | CLASS  | DIFF | STAT  |
> |----------------------------|--------|------|-------|
> | Zero-shot Direct Reasoning | 34.04  | 54   | 69.33 |
> | Chain of Thought (CoT)     | 23.40  | 40   | 65.33 |
> | **Our Method**             | 34.04  | 48   | 68    |
>
> **Key insights:**
> 1. ChemAgent consistently outperforms CoT, demonstrating its flexibility in enhancing stepwise reasoning across domains.
> 2. However, **direct reasoning** performed best, echoing observations in the Table 3 of the original SciBench paper, which noted degraded performance when using tools like Python for some LLMs (GPT-4-Turbo for example).
> 3. This may indicates that distilled models (like GPT-4omini) may suffer from reduced reasoning and coding capabilities compared to GPT-4.
>
> ### **Q4. Clarify sources and methodologies in Table 1**
>
> Thank you for pointing out this oversight. In our revised submission:
>
> - **Few-shot + Python** results were sourced from SciBench's benchmark paper (Table 3).
> - **Few-shot Direct Reasoning** results were obtained from the StructChem paper.
>
> We have updated the table's descriptions to accurately reflect these sources and methodologies, ensuring transparency.
>
> ### **Q5. Rationale for chosen benchmarks**
>
> We selected **SciBench** for three key reasons:
> 1. **Development Set Advantage**: SciBench provides a detailed **dev set** with clear solutions, streamlining our **library construction** phase.
> 2. **Comparative Baseline**: StructChem, a state-of-the-art method, is benchmarked on SciBench, providing a direct comparison.
> 3. **Recency**: SciBench was one of the most comprehensive and up-to-date benchmarks available when we began this research.
>
> ### **Q6. Performance on ChemBench**
>
> We evaluated ChemAgent on **ChemBench's physical chemistry subset** using **GPT-4o**.
>
> Results are as follows:
>
> | Method                     | Physical Chemistry |
> |----------------------------|--------------------|
> | Zero-shot Direct Reasoning | 33.33%            |
> | **ChemAgent**              | **83.33%**        |
>
> ChemAgent demonstrated significant improvements, further validating its effectiveness. We have included a detailed example trajectory from ChemBench in **Appendix H**.
>
> ### **Q7. Predefined templates vs. ChemAgent's approach**
> Thank you for offering the valuable suggestions about our mistake on describing previous methods. We have modified the paper according to your advices.
>
> About "predined templates": While ChemAgent uses predefined prompts to standardize task formats, it differs from template-based methods in key ways:
> 1. **Agentic Workflow**: ChemAgent dynamically adjusts the number and structure of subtasks based on task complexity, unlike rigid pre-designed templates.
> 2. **Iterative Refinement**: Through evaluation and refinement modules, ChemAgent can backtrack and revise strategies autonomously.
>
> ### **Q8. Missing citations and writing improvements**
>
> We have addressed the issues you raised:
> 1. **Citations**: Relevant works (e.g., [1-5], [7-11]) have been incorporated into the revised manuscript.
> 2. **Writing**: Ambiguous phrases (e.g., "predefined templates") and unclear sections (e.g., Table 1 descriptions) have been clarified. Changes are highlighted in red in the updated paper.

---

> ### Author Response · Authors · 2024-11-24
> **Response to Reviewer h1D2**
>
> ### **Q9. Ablation studies on the model used in the evaluation and refinement module**
>
> In **Appendix B, Table 4**, the last row demonstrates that using GPT-4 to evaluate ChemAgent's solutions and refining with GPT-3.5 outperforms the approach of using GPT-3.5 for both evaluation and refinement. However, in GPT-3.5 experiments, it still underperforms compared to ChemAgent operating without the evaluation and refinement module.
>
> This experiment highlights that the effectiveness of the evaluation and refinement module is highly dependent on the self-correction capabilities of the refinement model.
>
> Due to budget and computational resources constraints, we did not conduct further ablation studies to evaluate the impact of model selection on the performance of the evaluation and refinement module.

---

> > ### Comment · Reviewer_h1D2 · 2024-11-26
> >
> > Thank you so much for the detailed responses. Regarding the chemistry-specificity of your work, I would still argue that the only thing making it chemistry specific is your choice of tasks, but there's nothing specific go that field really. This is still fine though, but I think you should further justify this in the manuscript.
> >
> > I understand that changing the task without changing the stored memories would degrade performance, this is very clear. What I argue is that if you change the task (e.g. to physics or math) you would have to "retrain" the memory modules. Could you add some additional insights into how the system would change architecturally upon changing subjects?
> >
> > Regarding Q3, for these experiments did you retrain the memory modules? Can you please clarify this?
> >
> > For Q6, how do these 2 methods compare woth the other methods evaluated in the chembench paper? This is very good that you added these experiments, but the other results would be great to add for context.

---

> ### Author Response · Authors · 2024-11-27
> **Response to Reviewer h1D2**
>
> Thank you for kindly pointing out these issues.
>
> **Generalizability:**
>
> We sincerely appreciate your insightful feedback. In response, we have expanded our discussion on the generalizability of our method in the Conclusion and Appendix E.Limitation of the revised paper.
>
> We would like to clarify that when changing domains, the core structure of our framework remains unchanged—only the content of the memory library is modified. However, during the experiments presented in Table 2 (Section 4.1), we observed that for problems with shorter reasoning chains, removing the plan memory sometimes improved performance. Thus, when applying our framework to less complex domains, such as middle school mathematics, a potential modification to our architecture could involve removing the plan memory.
>
> **Regarding Q3 (Experiment on Math and Physics)**
>
> Yes, we retrained the memory modules for each experiment. In Q3's case, we did Library Construction for 3 times.
>
> **Compare with the other methods evaluated in the chembench paper?**
>
> The ChemBench paper evaluates only two methods beyond direct prompting with an LLM: (1) answering based on the log probabilities of the answer tokens, and (2) ReAct. The results for physical chemistry, as reported in their official repository, are as follows:
>
> | **Method**              | **Physical Chemistry** |
> |-------------------------|------------------------|
> | Zero-shot (GPT-4o)       | 50.00%                 |
> | ReAct (GPT-4o)           | 33.33%                 |
> | log_p (GPT-4o)           | 66.67%                 |
> | **ChemAgent** (GPT-4o)   | **83.33%**             |
>
> These results are directly sourced from the ChemBench GitHub repository. (Note: The performance of zero-shot direct reasoning using GPT-4o in the first row has been updated from 33.33% to 50.00% compared to our previous response, as we initially tested it ourselves and obtained 33.33%. The value in the ChemBench repository, however, is 50.00%.)
>
> As shown, our method outperforms all other approaches evaluated in the ChemBench paper.
>
> We hope this response clarify your questions and provide additional insights. Thank you again for helping us improve our work. Feel free to reach out for any further clarifications or discussions.

---

> > ### Comment · Reviewer_h1D2 · 2024-12-03
> >
> > Thank you very much for your detailed answers.
> > These results strengthen the work and have clarified some of my concerns.
> > I appreciate the author's effort into improving the work and I will upgrade my score to a 6 accordingly.
> > Thank you!

---

> > > ### Author Response · Authors · 2024-12-03
> > >
> > > We truly appreciate your dedication and the invaluable advice you've provided during this rebuttal period. Your insights have significantly impacted our work, and we are grateful for your support in helping us refine and strengthen the manuscript.

---

### Official Review · Reviewer_LmT4 · 2024-11-03

**Soundness:** 3
**Presentation:** 2
**Contribution:** 3
**Rating:** 6
**Confidence:** 4

**Summary:**

The paper introduces a framework ChemAgent aimed at enhancing chemical reasoning capabilities in large language models through a self-updating dynamic library. ChemAgent leverages a memory-based library system that decomposes complex chemical tasks into sub-tasks, storing solutions for future reference. The authors evaluate ChemAgent on four chemical reasoning datasets from SciBench, demonstrating that it significantly outperforms previous approaches.

**Strengths:**

1. The overall contribution of RAG+LLM with the application for chemistry calculation, is a rather novel contribution as an application.
2. This memory system, which includes planning, execution, and knowledge memory, is well-suited for tasks that require iterative problem-solving and cumulative knowledge retention.
3. The authors evaluated ChemAgent across multiple datasets and models and provided extensive error analysis and ablation study.

**Weaknesses:**

1. The main experiments use only one dataset (Sci-bench), and actually only the chemistry subset of this dataset. Given that chemistry reasoning is such a large topic with many datasets available, I would expect experiments on more datasets. Otherwise, the authors need to clearly explain why the other datasets are not suitable. Even if it’s the case that this dataset offers a unique benefit (e.g.,chemistry-related calculation, with labels ), other parts of the experiments can be performed on other datasets as well to strengthen the analysis.

2. While ChemAgent performs well against StructChem and other baselines, a direct comparison with more recent multi-modal or hybrid models (such as those integrating external tools like ChemCrow) would strengthen the case for ChemAgent’s utility across various LLM designs.[1]

[1].ChemCrow: Augmenting large-language models with chemistry tools

**Questions:**

1. During the library construction phase, the definition of "condition" is unclear, what kind of condition is extracted, and what is the criterion of "refine"? The authors are encouraged to clarify with an example.

2. Although the paper demonstrates strong performance in chemistry, how might the framework adapt to other scientific domains with similar reasoning demands, such as physics or computational biology? Are specific adjustments needed for these areas?

3. The evaluation and refinement module plays a significant role in performance. Could the authors elaborate on any observed limitations in this module, especially regarding instances where errors are detected but not effectively corrected?

4. How does ChemAgent compare with models that integrate external tools, such as ChemCrow, especially considering that code generation is employed to handle calculations in subtask answer generation? Are there established tools that could directly perform these calculations, and how do the authors assess the accuracy of the generated code?

---

> ### Author Response · Authors · 2024-11-24
> **Response to Reviewer LmT4**
>
> Thank you for your insightful review. Below, we address each of your questions in detail.
>
> ---
>
> ### **Q1. Clarify the library construction phase.**
>
> **A1.**
> Thank you for raising this question. We have added an example of the library construction phase in **Appendix H**. In summary:
> - **"Conditions"** refer to the given parameters in a problem, such as "temperature is 27°C" or "volume is 10L."
> - The **"refine" step** is essentially a mechanism designed to ensure that the extracted conditions are neither missing nor redundant.
>
> The specific prompt used for this step is detailed in **Appendix G.1**, under the section titled `REFLECT_PROMPT`.
>
> ---
>
> ### **Q2. Can our method be applied to other scientific domains with similar reasoning demands?**
>
> **A2.**
> Absolutely! Both you and reviewer h1D2 have highlighted this valuable point. Our method is indeed applicable to other scientific domains with similar reasoning requirements. Beyond minor adjustments to system instruction prompts and restarting the library construction phase, no significant modifications are necessary.
>
> We have included additional experiments and analyses in our **response to reviewer h1D2's Q2 and Q3**, demonstrating broader applicability and presenting interesting insights.
>
> ---
>
> ### **Q3. Elaborate on the observed limitations in the evaluation and refinement module.**
>
> **A3.**
> ChemAgent’s performance on the first test data (where 2θ is the correct answer) in ATKINS provides a representative example:
> ChemAgent is able to correctly report 2θ as the answer when the evaluation and refinement module is not activated, however, when the evaluation and refinement module is enabled, ChemAgent reports θ instead, which means an unnecessary refinement process is triggered.
>
> Additionally, as shown in **Figure 7 (Error 3)**, there are some types of errors hard to correct, even when the errors are detected. In cases like Error 3, the refinement module often selects other incorrect or irrelevant memories again.
>
> We attribute these limitations to:
> 1. The lack of relevant and useful memory.
> 2. The model's inherent difficulty in correcting its own reasoning errors without external guidance.
>
> ---
>
> ### **Q4. How does ChemAgent compare with models that integrate external tools, such as ChemCrow?**
>
> **A4.**
> Integrating specialized tools is an excellent suggestion. However, most tools integrated into ChemCrow (e.g., `Query_to_SMILES`) are not well-suited for tasks in **SciBench**.
>
> Our implementation, like ChemCrow, includes web search functionality. However:
> - Since the questions in **SciBench** are sourced from textbooks and generally searchable online, we did not evaluate this feature.
> - Additionally, web search introduces significant uncertainty into problem-solving.
>
> Nonetheless, we appreciate this suggestion as a promising direction for future research.
>
> ---
>
> ### **Q5. How do we ensure the correctness of the generated code?**
>
> **A5.**
> This is an excellent question. During initial experiments, we encountered issues with incorrect code generation. Our solution involves:
> 1. **Retry Mechanism:** If the generated code fails to execute, it is retried up to four times.
> 2. **Fallback Strategy:** If all retries fail, the model bypasses code generation and directly outputs its reasoning results.
>
> In practice:
> - Due to the simplicity of chemical codes, persistent code failures are rare.
> - For mathematical problems, particularly those involving equation solving, repeated failures were more common.
>
> We chose Python code over specialized computational tools to enhance the generalizability of our approach.
>
> ---
>
> ### **Q6. Why did we choose SciBench? And experiments on other datasets/benchmarks?**
>
> **A6.**
> We selected **SciBench** primarily for its well-defined step-by-step solutions in the dev set, which facilitated the initialization of our memory library without requiring additional manual effort.
>
> To evaluate broader applicability, we extended our library to test on **ChemBench's Physical Chemistry dataset**, achieving significant performance improvements:
>
> | **Method**              | **Physical Chemistry** |
> |--------------------------|------------------------|
> | Zero-shot (GPT-4o)        | 33.33%                |
> | **ChemAgent**            | **83.33%**            |
>
> These results underscore ChemAgent's effectiveness and generalizability across different datasets.
>
> ---
>
> We hope these responses clarify your questions and provide additional insights. Thank you again for your thoughtful feedback and for helping us improve our work. Please feel free to reach out for further clarifications or discussions.

---

> > ### Comment · Reviewer_LmT4 · 2024-11-26
> >
> > Thanks for the detailed response, I have no further concerns and will keep my accept score.

---

### Official Review · Reviewer_tXxm · 2024-11-04

**Soundness:** 3
**Presentation:** 2
**Contribution:** 2
**Rating:** 6
**Confidence:** 4

**Summary:**

The paper presents ChemAgent, a framework that enhances large language models (LLMs) in chemical reasoning tasks by dynamically self-updating memory library. The library is designed to retain knowledge and improve task performance over time by decomposing complex problems into smaller sub-tasks, storing solutions, and enabling efficient retrieval for future similar tasks. ChemAgent incorporates three types of memory: Planning Memory for high-level strategies, Execution Memory for specific solutions, and Knowledge Memory for core chemistry principles. Experimental results on four datasets from SciBench demonstrate that ChemAgent outperforms baselines, particularly in complex chemistry tasks.

**Strengths:**

* The three-part memory system (Planning, Execution, Knowledge) effectively models how LLMs can incrementally improve through task decomposition and retrieval. This approach emulates human memory retrieval, making it highly applicable to chemistry, where multi-step reasoning and recall of specific formulas or constants are frequently required.
* The experimental setup provides a solid justification for the additional memory components, clearly demonstrating the benefits of enhanced recall and problem-solving.

**Weaknesses:**

* As the library expands, retrieval efficiency and memory management may become bottlenecks. The framework’s performance could be compromised in environments with frequent memory updates and large datasets, highlighting a need for memory pruning or optimization strategies.
* More implementation details about the memory architecture would be beneficial, especially regarding the memory sizes and updating mechanisms. An analysis of the relative sizes of each memory type (Planning, Execution, Knowledge) would be insightful.

**Questions:**

* How does ChemAgent handle potential memory overflow? Is there a mechanism to dynamically optimize memory usage?
* Follow up on W2, what are the sizes of each memory module, and do they vary by type? If so, what are the considerations behind these differences?
* How long does it typically take to build a knowledge library that achieves reliable performance?
* Is there a mechanism to ensure the quality of data accumulated over time?

---

> ### Author Response · Authors · 2024-11-24
> **Response to Reviewer tXxm**
>
> We sincerely appreciate your insightful feedback and thoughtful questions about ChemAgent's memory management system. Please allow us to address your concerns comprehensively.
>
> ### **Q1. How to handle potential memory overflow? How do we dynamically optimize memory usage?**
>
> **A1.**
> Generally, we have not focused on controlling the size of the memory library for the following reasons:
> 1. Our work primarily emphasizes the effectiveness of self-updating memories in reasoning tasks.
> 2. The dataset we use (SciBench) is relatively small, making memory overflow unlikely.
>
> However, in Section 3.3 (page 8, paragraph 2: “To ensure accuracy and prevent target leakage, …, for any j < k.”), we discuss an example of managing memory usage. Specifically, we exclude certain memories during specific iterations by examining their similarity. For instance, in Section 3.3, memories with a similarity score higher than 0.88 with $\mathcal{P}_i$ are discarded. While this method controls memory usage, it does not directly optimize it.
>
> A straightforward solution to handle memory overflow would be to discard memories with high similarity to existing ones. However, this approach may negatively impact the performance of ChemAgent. Addressing this trade-off remains an open question and is worth further exploration. We have added this limitation in the **Limitations** section of our paper.
>
> ---
>
> ### **Q2. What are the sizes of each memory module, and do they vary by type? What are the considerations behind these differences?**
>
> **A2.**
> Yes, the sizes of memory modules vary by type. The size is influenced by whether ChemAgent is allowed to self-evolve during runtime (Section 3.3).
>
> 1. **Without self-evolution during runtime:**
>    The sizes of plan memory and execution memory primarily depend on the size of the development set for a given dataset. For example, using the ATKINS dataset:
>    - Execution memory: 8 MB
>    - Plan memory: 3 MB
>    The dev-set size is 16. As described in Section 2.4 and Figure 2, incorrect solutions are discarded during the library construction phase. Thus, datasets with the same dev-set size may have slightly different memory module sizes due to differences in discarded solutions.
>
>    The memory statistics for various datasets are summarized below:
>
>    | **Dataset**       | **Dev-set size** | **Plan Memory** | **Execution Memory** |
>    |-------------------|------------------|-----------------|----------------------|
>    | **ATKINS**        | 16               | 2.9 MB          | 8.1 MB              |
>    | **CHEMMC**        | 9                | 1.5 MB          | 5.3 MB              |
>    | **MATTER**        | 10               | 6.3 MB          | 4.1 MB              |
>    | **QUAN**          | 8                | 1.7 MB          | 3.7 MB              |
>
> 2. **With self-evolution during runtime:**
>    In this case, the sizes of memory modules are approximately proportional to the number of tasks ChemAgent has encountered.
>
> For both settings, the size of **knowledge memory** is dynamic because we do not store it. Instead, ChemAgent dynamically generates knowledge memory at runtime based on the specific problem it is solving.
>
> There are no manually designed considerations behind the differences in memory module sizes. Execution memory is typically larger than plan memory because a single plan may include multiple sub-tasks, each contributing to execution memory.
>
> ---
>
> ### **Q3. How long does it typically take to build a knowledge library that achieves reliable performance?**
>
> **A3.**
> The time required to build the library depends on the size of the dev-set and the inference speed of the backbone LLM. For example, with the ATKINS dataset and GPT-4:
> - Approximately 6~7 hours are needed, mainly due to the latency of the GPT-4 API.
>
> Using local models can significantly reduce this time.
>
> The minimum dev-set size required to achieve reliable performance remains an open question and is outside the primary scope of our research.
>
> ---
>
> ### **Q4. Is there a mechanism to ensure the quality of data accumulated over time?**
>
> **A4.**
> Yes, there are mechanisms in place to ensure data quality:
> 1. **During the Library Construction phase:** Memories that the LLM evaluates as incorrect trajectories are discarded.
> 2. **During runtime experiments (e.g., Section 3.3):** Only memories derived from correctly solved tasks are inserted into the library.
>
> ---
>
> We hope these explanations clarify our approach and address your concerns. Please feel free to reach out for further discussion or clarifications.

---

> > ### Comment · Reviewer_tXxm · 2024-11-25
> >
> > Thanks for the additional clarification and discussions! I am maintaining my current score.

---

### Official Review · Reviewer_ea4h · 2024-11-04

**Soundness:** 2
**Presentation:** 2
**Contribution:** 2
**Rating:** 5
**Confidence:** 4

**Summary:**

The paper introduces a novel framework named *ChemAgent* designed to enhance the performance of large language models (LLMs) in complex chemical reasoning tasks. This framework overcomes the typical limitations faced by LLMs in accurately handling domain-specific formulas and reasoning steps.

**Key contributions of the paper include:**
1. **Dynamic Library System**: ChemAgent incorporates a self-updating "library" system that decomposes complex chemical problems into sub-tasks, storing these alongside their solutions for future reference. This approach allows the system to retrieve and refine relevant information when tackling new problems.
2. **Memory Components**: The framework introduces three types of memory:
   - **Planning Memory** for high-level strategies.
   - **Execution Memory** for detailed problem-solving steps.
   - **Knowledge Memory** for basic chemistry principles.
3. **Iterative Problem Solving**: ChemAgent's iterative learning enables LLMs to improve their accuracy over time, enhancing performance through experience.
4. **Performance Gains**: The framework demonstrated up to a 46% improvement in chemical reasoning accuracy on SciBench datasets, significantly outperforming existing methods such as StructChem.

The authors conclude that ChemAgent holds promise for applications in fields like drug discovery and materials science, and they provide their code for further research and development.

**Strengths:**

1. **Self-updating dynamic memory library system**: ChemAgent improves the chemical reasoning capabilities of large language models (LLM) by introducing a self-updating "library" system. The library improves reasoning and solution accuracy by decomposing complex chemical problems into subtasks and storing these tasks and corresponding solutions in memory components, allowing models to retrieve and leverage the past when solving new problems.

2. **Performance improvement**: Experimental results show that ChemAgent achieves up to 46% performance improvement in chemical reasoning tasks on the SciBench dataset, significantly outperforming existing solutions such as StructChem. Especially when using more powerful models such as GPT-4, ChemAgent has demonstrated the ability to continuously learn and improve through its memory system and evaluation improvement module.

**Weaknesses:**

1. **Clarification of Innovation and Contribution**: ChemAgent introduces a dynamic memory updating mechanism, which, while innovative, relies on existing techniques and frameworks, such as prompt engineering and the combination of pre-trained models, to enhance the performance of large language models (LLMs). While the self-improvement capabilities of the dynamic library system are noteworthy, the approach depends primarily on prompt design to manage LLM interactions and context retrieval, an area that has been extensively explored. Therefore, we suggest authors further emphasize the distinctiveness and novelty of ChemAgent’s integration of prompt design and memory modules to help better differentiate its contributions from existing works.
2. **Generalizability and Comprehensiveness of Experimental Evaluation**: While the SciBench dataset provides a rigorous benchmark for evaluation, the current experiments are focused on specific chemical reasoning tasks. Expanding the validation to include more diverse datasets or real-world scenarios like optimization of chemical experiment parameters would strengthen the argument for ChemAgent's broader applicability. We recommend considering a wider range of more challenging evaluation tasks in future work to comprehensively assess the system's performance across a variety of chemical or scientific problems.

**Questions:**

### **Clarification of Innovation and Contribution**

**Questions:**

- Could the authors provide a more detailed explanation of how ChemAgent’s integration of dynamic memory updating and prompt engineering differs from existing approaches? Specifically, how does ChemAgent’s memory updating mechanism contribute novel insights or advancements in the field of LLMs, beyond the well-explored techniques of prompt design and memory retrieval?
- Could the authors elaborate on any unique aspects of ChemAgent's architecture or framework that differentiate it from similar systems in the literature? Are there any specific design choices made to optimize memory retrieval or self-improvement that haven’t been widely explored in other works?

**Suggestions:**

- We suggest that the authors further emphasize the novelty of ChemAgent's integration of prompt design and memory modules. A comparison to existing memory-enhanced models in the LLM space could help highlight ChemAgent’s unique contribution.
- We recommend that the authors provide a more thorough description of how the dynamic memory updating mechanism operates in real-time and how it evolves through interactions. This would help clarify the distinction between ChemAgent and previous works focused on prompt engineering or static memory updates.

---

### **Generalizability and Comprehensiveness of Experimental Evaluation**

**Questions:**

- While the use of the SciBench dataset is commendable, could the authors clarify if the results obtained are robust across a wider range of chemical tasks? Have the authors considered evaluating ChemAgent on real-world chemical challenging reasoning tasks, such as optimizing experimental parameters or suggesting experimental protocols?
- How does ChemAgent perform when applied to more complex, interdisciplinary problems, such as tasks that require domain-specific knowledge integration (e.g., chemical engineering or materials science)?

**Suggestions:**

- We suggest that the authors expand the experimental evaluation to include a broader range of datasets, including those that involve real-world scientific challenges. This would provide a clearer picture of the system's generalizability and its potential for application in diverse chemical and scientific domains.
- We recommend that the authors explore performance on tasks with varying levels of difficulty, such as those requiring multi-step reasoning or integration of data from multiple sources, to better assess ChemAgent's capabilities across different domains.

---

> ### Author Response · Authors · 2024-11-24
> **Response to Reviewer ea4h**
>
> We thank the reviewer for this insightful question. We want to clarify several key differences between ChemAgent and existing approaches:
>
> Unlike existing prompting methods like Chain-of-Thought (CoT), Tree of Thoughts (ToT), or Buffer of Thoughts (BoT) that focus on improving reasoning through better prompting structures, ChemAgent introduces a novel dynamic memory system specifically designed for chemical reasoning. While methods like ToT maintain a tree structure for exploration, our approach maintains a structured knowledge repository that evolves with each problem-solving experience.
>
> Compared to other memory-augmented approaches, ChemAgent incorporates domain-specific structure through its three distinct memory types (Planning Memory, Execution Memory, and Knowledge Memory). This specialized architecture allows for:
> - Hierarchical decomposition of chemical problems
> - Storage of verified solution strategies
> - Dynamic updating of chemical domain knowledge
>
> We have added a new comparison table that highlights these differences:
>
> | Method          | Memory Type    | Update Mechanism    | Domain Adaptation        |
> |------------------|----------------|---------------------|--------------------------|
> | ToT [14]        | Tree Search    | Static              | General                  |
> | GoT [17]        | Graph          | Static              | General                  |
> | PAL [10]        | None           | None                | Programming              |
> | ChemAgent       | Hierarchical   | Dynamic+Verified    | Chemistry       |
>
> ---
>
> > Could the authors elaborate on any unique aspects of ChemAgent's architecture or framework that differentiate it from similar systems in the literature?
>
> The key architectural innovations of ChemAgent include:
> - Domain-specific memory structure: Unlike general-purpose systems, ChemAgent's memory components are specifically designed for chemical reasoning:
>   - Planning Memory captures high-level chemical problem-solving strategies.
>   - Execution Memory stores detailed solution steps with chemical formulas and calculations.
>   - Knowledge Memory maintains fundamental chemistry principles.
>
> - Runtime memory updates: We have implemented a novel verification mechanism that ensures correct and useful information is added to the memory pool. Our experiments (Section 3.3) show this improves accuracy by 7% compared to systems without updating.
>
> ---
>
> > While the use of the SciBench dataset is commendable, could the authors clarify if the results obtained are robust across a wider range of chemical tasks? Have the authors considered evaluating ChemAgent on real-world chemical challenging reasoning tasks, such as optimizing experimental parameters or suggesting experimental protocols?
>
> Thank you for this suggestion. We acknowledge this limitation. However, we note that comprehensive evaluation on experimental optimization tasks is currently limited by the lack of standardized benchmarks. This presents an opportunity for future work in creating such datasets.
>
> The importance of chemical reasoning tasks is highlighted by recent works in drug discovery and materials science (citations added). Our framework provides a foundation for tackling these complex tasks through its ability to learn and refine chemical problem-solving strategies.
>
> ---
>
> > The authors should explore performance on tasks with varying levels of difficulty, such as those requiring multi-step reasoning or integration of data from multiple sources.
>
> We agree with this suggestion and have added new experiments on other benchmarks (ChemBench's Physical Chemistry sub-dataset).
>
>
> | **Method**              | **Physical Chemistry** |
> |--------------------------|------------------------|
> | Zero-shot (GPT-4o)       | 33.33%                |
> | **ChemAgent**            | **83.33%**            |
>
> These results demonstrate our system's robustness across varying difficulty levels and different datasets while highlighting areas for future improvement.

---

> > ### Author Response · Authors · 2024-11-26
> > **Response to Reviewer ea4h**
> >
> > ### **Regarding Novelty**
> > Generally, to our knowledge, ChemAgent is the first effort to integrate various memory components into a cohesive framework that allows dynamic updates, leading to gradual improvements in performance. Furthermore, our work specifically targets problems in the Chemistry domain, which demand higher reasoning capabilities compared to the everyday tasks studied by previous agent researches. In this hard scientific context, the question of "how to enhance an agent's reasoning ability" remains largely underexplored.
> >
> > Thank you once again for your thoughtful comments. If you have further questions, we are willing to provide additional clarification.

---

### Author Response · Authors · 2024-12-02

We thank all the reviewers for their time and constructive comments. We would like to know whether our responses have fully addressed your concerns. Please feel free to comment if there are any further confusions.

---

### Meta-Review · Area_Chair_B1yU · 2024-12-21

**Metareview:**

This paper proposes a new framework named ChemAgent to enhance the performance of LLMs for chemical language models. Their main idea is to introduce three types of self-updating (planning, knowledge, and execution) memories that aid the LLM's chemical reasoning ability.

This paper is well-executed and provides meaningful and convincing improvement to chemical reasoning ability of LLMs.

While the method is not entirely novel, there exists some new components. The dynamically updated memory idea is not entirely restricted to the application to chemical reasoning ability and the idea can be generalized.

Overall, I recommend acceptance for this paper. I think this provides a solid and meaningful contribution to chemical LLMs, which is an important application of LLMs.

**Additional Comments On Reviewer Discussion:**

Reviewer ea4h raised concerns on novelty of the work, but the concerns were not very specific and the reviewer did not respond to the rebuttal. From my assessment, this concerns are well addressed.

Other concerns from the reviewers were addressed, as acknowledged by the reviewers.

---

### Decision · Program_Chairs · 2025-01-22

Accept (Poster)